# BetterBench: Assessing AI Benchmarks, Uncovering Issues, and Establishing Best Practices

**Anka Reuel** *
Stanford University

**Amelia Hardy**\*
Stanford University

**Chandler Smith**
Northeastern University

**Max Lamparth**
Stanford University

**Malcolm Hardy**
Stanford University

**Mykel J. Kochenderfer**
Stanford University

## Abstract

AI models are increasingly prevalent in high-stakes environments, necessitating thorough assessment of their capabilities and risks. Benchmarks are popular for measuring these attributes and for comparing model performance, tracking progress, and identifying weaknesses in foundation and non-foundation models. They can inform model selection for downstream tasks and influence policy initiatives. However, not all benchmarks are the same: their quality depends on their design and usability. In this paper, we develop an assessment framework considering 46 best practices across an AI benchmark's lifecycle and evaluate 24 AI benchmarks against it. We find that there exist large quality differences and that commonly used benchmarks suffer from significant issues. We further find that most benchmarks do not report statistical significance of their results nor allow for their results to be easily replicated. To support benchmark developers in aligning with best practices, we provide a checklist for minimum quality assurance based on our assessment. We also develop a living repository of benchmark assessments to support benchmark comparability, accessible at betterbench.stanford.edu.

## 1 Introduction

AI systems are rapidly advancing and proliferating [58]. The increasing integration of AI, and in particular foundation models (FMs) [14], into decision-making systems has significantly amplified its impact and has showcased both benefits [9, 39, 57, 66] and risks [2, 75, 44, 86, 45, 30, 70]. Given the importance of correctly assessing a model's capabilities and potential harms, AI evaluation is an essential discipline [15]. Current evaluation approaches include both internally (e.g., private testing on proprietary data) and externally developed techniques (e.g., scoring on public benchmarks) [74, 27, 73, 48, 32].

Following the work of [67], we define a benchmark "as a particular combination of a dataset or sets of datasets [...], and a metric, conceptualized as representing one or more specific tasks or sets of abilities, picked up by a community of researchers as a shared framework for the comparison of methods" [67]. Using benchmarks to facilitate comparison, measure performance, track progress, and identify weaknesses has become a standard practice. For example, benchmarks are widely used by model developers to report performance and compare models upon release [3, 8], and as part of policy initiatives to support third-party model evaluations, such as as part of the UK AI Safety Institute's

---

*(*) denotes equal contribution. Corrspeonding authors: anka.reuel@stanford.edu, ahardy@stanford.edu

Submitted to the 38th Conference on Neural Information Processing Systems (NeurIPS 2024) Track on Datasets and Benchmarks. Do not distribute.

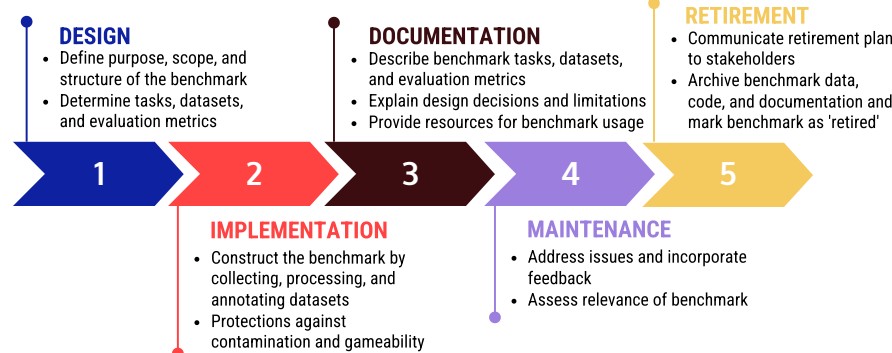

Figure 1: Five stages of the benchmark lifecycle. A detailed description can be found in App. C.

*Inspect* framework for evaluating large language models (LLMs) [81] or Article 51 of the EU AI Act [1]. However, the fidelity of this approach depends entirely on the benchmarks' quality, where we define a *high-quality* benchmark as one that is interpretable, clear about its intended purpose and scope, and that is usable. To date, no structured assessment for the quality of AI benchmarks, including both FM and non-FM benchmarks, has been published, and no comparative analysis has been conducted to understand quality differences between widely used AI benchmarks. To address these gaps, our paper:

- Presents a novel AI benchmark assessment framework evaluating the quality of AI benchmarks based on 46 criteria derived from expert interviews and domain literature
- Scores 16 foundation model (FM) and 8 non-FM benchmarks (full list in App. D), finding quality differences across both categories
- Provides insights into prevalent issues in current AI benchmarking practices based on our assessment
- Creates a checklist for minimum quality assurance to support benchmark developers in aligning with best practices
- Makes available a living repository[2] of benchmark assessments for users to analyze benchmarks' quality and appropriateness for their usage contexts.

We structure the paper as follows: Sec. 2 explores benchmarking in AI and other fields. Sec. 3 describes our assessment development, which combined literature and expert interviews, and details our benchmark scoring procedure. Sec. 4 presents our framework's criteria, focusing on aspects under developers' control to promote better benchmarks. Sec. 5 lists additional context-dependent design considerations. Sec. 6 reports findings from applying our framework to 24 benchmarks. Finally, Sec. 7 and Sec. 8 explore implications for future evaluations and discuss our work's scope and limitations. We further outline open challenges with AI benchmarking in App. A, involved stakeholders in App. B, and the AI benchmark lifecycle in App. C.

## 2 Related Work

### 2.1 AI Benchmarking Practices and Challenges

Our literature review of AI benchmarking practices identifies two primary concerns: what a benchmark measures and how this measurement is used. Regarding what a benchmark measures, [59] find that current benchmarks for LLMs are insufficient for assessing these models capabilities. A frequent concern in this context is the validity of evaluations [54, 76, 67]. Similarly, [62] finds

---

[2]https://betterbench.stanford.edu

that the rapid advancement of AI models threatens benchmarks' utility, as a large fraction of these evaluations are near saturation. [83] and [49] both address the narrow scope of existing benchmarks, with [49] advocating for approaches intended to reduce the socio-technical gap that exists between the capabilities that benchmarks are able to measure and the ability of models to meet user needs in downstream applications. With respect to how evaluations are used, [67] critiques the tendency of AI practitioners to overgeneralize benchmark results, highlighting how these scores present an inherently reductive view of model performance.

In addition, the community has also recognized the importance of data curation and documentation in the context of evaluations. [65] put forth the idea of data cards as standardized documentation framework for datasets and [12] develop a framework and checklist for best practices in data curation. Finally, the FAIR principles [87] outline best practices for digital data access, based on the principles of *Findability*, *Accessibility*, *Interoperability*, and *Reuse*. While these efforts support the adoption of best practices in the context of data, they are insufficient for assessing AI benchmarks, which extend data with infrastructure and evaluation methods, requiring additional guidelines to support the development of high-quality benchmarks and the decision-making of benchmark users.

Hence, our work builds on and expands these guidelines, with the aim of advancing the analysis of AI benchmarking by presenting a first-of-its-kind framework for the assessment of both foundation model and non-foundation model benchmarks. Unlike prior studies, such as [59] and [49], which focus on identifying limitations in limited contexts and scopes, our approach offers practical tools, empowering developers to address shortcomings and directly enhance benchmark quality: Our assessment spans a wider range of criteria across the benchmark lifecycle, from design (e.g., have domain experts been involved in the development?) to implementation (e.g., is the evaluation script available?), documentation (e.g., is the applicable license specified?), and maintenance (e.g., is a feedback channel available for users?). We give an overview of all our criteria in Sec. 4 and explain, justify, and provide scoring details for each criterion in App. K. We further provide a checklist of best practices derived from our analysis (App. J), offering guidance for improving AI benchmarks, rather than merely highlighting issues.

## 2.2 Benchmarking Best Practices in Other Fields

Our work is informed by benchmarking practices from fields beyond AI, ranging from transistor hardware [18] to environmental quality [16] to bioinformatics [7], and we identify common themes regarding what constitutes an effective benchmark. Where applicable, we incorporate these best practices into our assessment (Sec. 4):

**Designing for downstream utility.** Many of the papers reviewed discuss the importance of a benchmark's tasks being designed with real world applications in mind. [16] considers the best benchmarks to be situation-specific, [24] defines an ideal test set as one which reflects real world data, [7] proposes that benchmarks should be adapted to their intended applications, and [25] suggests that benchmarks be designed to fit the diversity of downstream use cases. [77] emphasizes the importance of guaranteeing that tested methods only use information available in a practical setting and recommends checking that a benchmark simulates the envisioned usage.

**Ensuring validity.** A frequent concern with benchmarking is the validity of evaluations [54, 76, 67]. In educational testing, [60] outline a framework to ensure validity by providing guidelines for effective evidence collection. [22] outline what and how evidence can be collected and how it should be interpreted for tests "of attributes for which there is no adequate criterion" [22]. Measures that are used in other fields further include choosing a large test set to promote the statistical significance of results [77] and updating a benchmark over time to prevent developers from overfitting it [7]. [7] also notes that the methods or approaches being evaluated should not be used to create the gold standard dataset.

**Prioritizing score interpretability.** [7] highlights that benchmarks are particularly important when a wide variety of tools are available and it is difficult for non-specialists to distinguish between them. Interpretability is important in not only selecting tools, but also deciding between benchmarks

themselves. Effective benchmarks must provide transparent information regarding the procedural details of their experiments [18] and goals of the evaluation [10]. They should clearly describe the benchmark's purpose and scope, as these are fundamental to its design and implementation [85]. Regarding scope, [16] states that for environmental quality applications, benchmarks should never be the basis of final decisions. With this in mind, they identify misleading benchmarks as the worst-case scenario. Furthermore, they state that a benchmark should not present its results as absolutes, instead ensuring that its evaluations are understandable inputs for decision makers [16].

**Guaranteeing accessibility.** A good benchmark is easy to obtain and use [7, 77, 25, 10]. If a benchmark is run computationally, then its data and scripts must be available for results to be reproducible [77, 25, 10].

## 3 Methodology

Our benchmark assessment consists of 46 criteria based on our literature review and interviews with five primary groups of stakeholders. These groups, who also present the user personas of our assessment, are described in detail in App. B. Through our interview process, we defined a five-stage benchmark lifecycle and identified objectives along it. In this section, we discuss our methodology for identifying stakeholders, developing criteria, and assessing benchmarks. A detailed flow diagram of our methodology can be found in App. H.

**Step 1: Mapping the space.** Initially, we surveyed the existing benchmark landscape (Sec. 2). Based on this review, we identified five stakeholder groups who present the user personas of our assessment (App. B). To understand their objectives with respect to benchmarking, we conducted unstructured interviews with representatives of all stakeholder groups, including 20+ policymakers, model developers, benchmark developers, model users, and AI researchers. During this process, we developed a five-stage model of the benchmark lifecycle (Fig. 5 and App. C) and mapped both the benchmarking objectives of the stakeholders and their communicated use cases for a benchmark assessment (App. B).

**Step 2: Translation to criteria.** Based on Step 1, we identified tasks and objectives for each stage of the AI benchmark lifecycle and translated them into concrete criteria. We categorized these as: (a) criteria controlled by the benchmark developer where the authors and interviewees reached a normative consensus, (b) criteria controlled by the benchmark developer but context-dependent, difficult for an external party to assess, or both and (c) aspects either outside the benchmark developer's control or requiring further research. The assessment in Sec. 4 is limited to category (a) criteria. We cover considerations in (b) in Sec. 5, and those in (c) in App. A.

**Step 3: Validating the assessment.** Initially, three authors independently scored the same benchmark to calibrate the assessment and identify potential misinterpretations of the criteria. We adapted and clarified scoring guidelines (App. K) to address differing interpretations and uncertainties. To validate our assessment, we shared it with members of all stakeholder groups and revised it based on their feedback. Finally, we verified that our assessment, which in itself can be considered a benchmark, met all of our defined criteria, where applicable (App. J.2).

**Step 4: Structuring the assessment.** We evaluated 16 FM and 8 non-FM benchmarks. We prioritized commonly used benchmarks, such as those that were recently reported by model developers [8, 3] and aim to expand the number of assessed benchmarks continuously on our website *betterbench.stanford.edu*. Since our assessment considers varying information sources (official websites, papers, GitHub repositories published by the benchmark developers[3]) that do not follow a standard structure, we manually evaluated all benchmarks. At least two authors independently reviewed each benchmark. They subsequently had to reach a consensus on the final score and a third reviewer could be called to make the final decision if a consensus could not be reached (this case did not occur).

---

[3]We do not consider third-party information that was not released by the benchmark developers themselves.

**Step 5: Scoring.** We scored benchmarks on a discrete 0/5/10/15-point scale for each criterion: 15 for fully meeting, 10 for partially meeting, 5 for mentioning without fulfilling, and 0 for neither referencing nor satisfying the criterion. Average scores were calculated for each benchmark lifecycle stage (design, implementation, documentation, and maintenance). An aggregate usability score, representing the weighted average of the implementation, documentation, and maintenance scores, was also introduced (see App. G for scoring details). We consider a mean score of 10 or higher to indicate a reasonably good benchmark for each aggregated scoring category, as it signifies that, on average, the benchmark at least partially fulfills all assessment criteria within the respective category.

**Step 6: Platform for continuous updates.** Finally, we develop a supplementary website[4] to continuously publish assessment results using the scoring methodology in App. G, given the rapid development of new AI benchmarks. The website includes a community feedback channel for submitting new AI benchmarks and correcting previously posted scores if benchmarks are updated or stakeholders disagree with our evaluation. This provides benchmark users with an accessible, up-to-date database of existing benchmarks and their quality, enabling quick analysis of the most suitable benchmark for their application context.

# 4 Assessment Criteria

We separate our assessment criteria according to the phase of the benchmark lifecycle during which they would be fulfilled. Although the retirement stage is within the developer's control, we do not include specific criteria for this phase within the current framework, because we cannot assess the retirement of active benchmarks. App. K contains full explanations, justifications, and scoring guidelines for each of the 46 criteria.

## 4.1 Benchmark Design

**Design Criteria**

1. Tested capability, characteristic, or concept is defined
2. How tested capability or concept translates to benchmark task is described
3. Domain experts are involved
4. Domain literature is integrated
5. Use cases or user personas are described
6. Differences to related benchmarks are explained
7. Input sensitivity is addressed
8. Has validated automatic evaluation
9. How benchmark score should or shouldn't be interpreted or used is described
10. How knowing about the tested concept is helpful in the real world is described
11. Informed performance metric choice
12. Metric floors and ceilings are included
13. Human performance level is included
14. Random performance level is included

Figure 2: Overview of assessment criteria for the benchmark design stage.

Benchmarks should clearly describe their goals and scope [85, 10, 54]. This includes defining the tested capability or characteristic, describing how the tested capability translates to the benchmark task, and stating how knowing about the tested concept is helpful in real-world applications [54]. These design choices should be informed by considering use cases and user personas for the benchmark, involving domain experts, and integrating domain literature [82]. Clearly stating how the benchmark is different from related existing AI benchmarks is necessary to help benchmark users decide the applicability of a benchmark to their use case. A benchmark's measurements must be interpretable [16], which requires an informed choice of performance metric(s) and a description of how the benchmark score should or shouldn't be interpreted [48]. Including floors, ceilings, human performance levels, and random performance levels for the chosen metric(s) further assists users in understanding a model's score [34]. If addressing input sensitivity and providing a validated automatic evaluation are possible, these measures enhance a benchmark's robustness and accessibility [34].

---

## 4.2 Benchmark Implementation

**Implementation Criteria**

1. Evaluation code is available
2. Evaluation data or generation mechanism is accessible
3. Evaluation of models via API is supported
4. Evaluation of local models is supported
5. Globally unique identifier or encryption of evaluation instances is added
6. Task to identify if model has been trained on benchmark data is included
7. Script to replicate results is explicitly included
8. Statistical significance or uncertainty quantification of benchmark results is reported
9. Need for warnings for sensitive/harmful content is assessed
10. Build status is implemented
11. Release requirements are specified

Figure 3: Overview of assessment criteria for the benchmark implementation stage.

Criteria in the implementation stage focus on the availability of necessary code and infrastructure and the inclusion of key engineering features. To ensure reproducibility and scrutiny [77, 25, 10], a benchmark should provide working evaluation code, and make its evaluation data, prompts, or dynamic test environment accessible. A script should be available to replicate initial published results. In domains where models are often accessed via API, such as NLP, an ideal benchmark supports the evaluation of both API-based and local models. A benchmark can minimize the risks of contamination and gamification by including a globally unique identifier or encrypting evaluation instances. This is especially important for testing models that rely on web-scraped training data. Including a *training_on_test_set* task allows determining whether a model's training data included benchmark examples [74]. As an additional measure, specifying clear release requirements informs users how to preserve the integrity of test results [6].

## 4.3 Benchmark Documentation

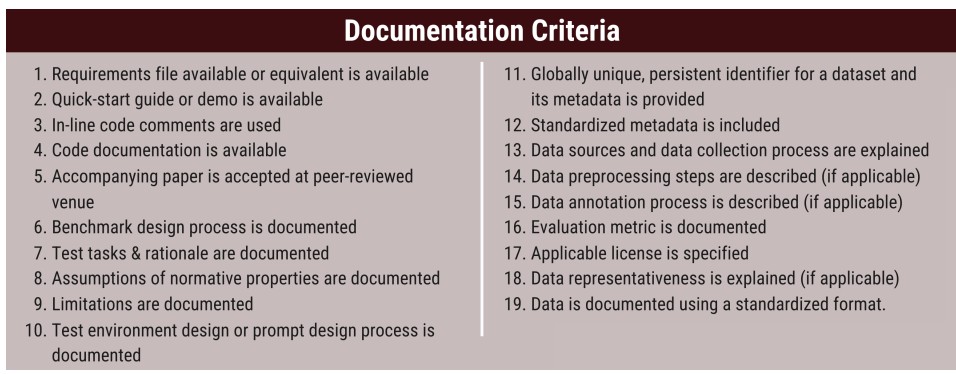

**Documentation Criteria**

1. Requirements file available or equivalent is available
2. Quick-start guide or demo is available
3. In-line code comments are used
4. Code documentation is available
5. Accompanying paper is accepted at peer-reviewed venue
6. Benchmark design process is documented
7. Test tasks & rationale are documented
8. Assumptions of normative properties are documented
9. Limitations are documented
10. Test environment design or prompt design process is documented
11. Globally unique, persistent identifier for a dataset and its metadata is provided
12. Standardized metadata is included
13. Data sources and data collection process are explained
14. Data preprocessing steps are described (if applicable)
15. Data annotation process is described (if applicable)
16. Evaluation metric is documented
17. Applicable license is specified
18. Data representativeness is explained (if applicable)
19. Data is documented using a standardized format.

Figure 4: Overview of assessment criteria for the benchmark documentation stage.

Providing comprehensive and accessible documentation is crucial for the practicability and interpretation of benchmarks [18]. Key information about a benchmark should be readily available and include documentation of benchmark construction processes [54], data collection [87] or test environment design, and its test tasks and their rationale [54]. Clearly documenting evaluation metric(s) and reporting the statistical significance of results is necessary so that users can understand a benchmark's actual signal [4]. To provide context and prevent misinterpretation, developers should document normative assumptions about benchmark properties and discuss the limitations of their benchmark. A benchmark's codebase should contain a requirements file, a quick-start guide or demo code, a description of code file structure and contents, and in-line comments within all relevant files. Having a benchmark's paper accepted at a peer-reviewed venue signals external scrutiny and adherence to certain standards. Lastly, developers should specify the applicable license to provide legal clarity and enable, e.g., commercial use.

## 4.4 Benchmark Maintenance

| Maintenance Criteria | |
|---|---|
| 1. Code usability was checked within the last year
2. Maintained feedback channel for users is available | 3. Contact person is listed |

Figure 5: Overview of assessment criteria for the benchmark maintenance stage.

An optimally designed, implemented, and documented benchmark will cease to be useful if it is not maintained. Developers should regularly check code usability and maintain a feedback channel for users to report issues or suggest improvements. Providing contact details of a person responsible for the benchmark facilitates communication and support. Alternatively, if a benchmark is not maintained anymore, authors should include a corresponding statement indicating that the benchmark was retired in any official benchmark artefacts.

# 5 Other Design Considerations

This section presents design considerations for benchmark developers that were excluded from our assessment because their appropriateness is context-dependent, they are not easily verifiable, or both. Our aim with this list is to promote conscious design decisions regarding these considerations.

**General vs. specific benchmarks.** Benchmark developers must decide whether to prioritize general or abstract knowledge and skills or specific contexts and domains. Broad concept benchmarks may contribute to understanding foundational characteristics of models, but often face challenges in real-world applicability and reliable testing (see App. A).

**Detecting small improvements.** Benchmarks should be designed so that a 1% improvement can be reliably detected [34]. As [34] states, "the more difficult it is to detect small amounts of progress, the more difficult it becomes to make iterative progress on a benchmark." Practically, this is likely dependent on evaluation data size and task diversity.

**Multi-modal assessment.** As multi-modal models become increasingly common, benchmark developers may want to consider designing tasks to assess the capabilities they want to test across modalities. Additional design considerations for multi-modal assessments include the increased complexity of mapping a tested concept to different modalities and the different output formats of the tested models [91].

**Versioning.** Minor updates (e.g., removing faulty prompts) should be clearly indicated via *task versioning* [13]. Major updates require releasing new *benchmark versions*, as exemplified by the AgentBench v0.1 and v0.2 releases [52].

**Dynamic vs. static benchmarks.** Dynamic benchmarks may better address quick saturation (App. A) and contamination (App. A) issues but reduce result comparability and are easier to implement for some tasks (e.g., adding numbers) than others. Static benchmarks, on the other hand, tend to suffer from the issues outlined above.

**Gameability.** An ideal benchmark is resilient to attempts to boost task performance without improving the fundamental capability being tested [7]. Existing benchmarks have been shown to be vulnerable to manipulation [6]. Specific guidelines have been proposed to prevent cheating and ensure evaluations reflect genuine model performance [94].

**Positionality statement.** Positionality statements[5] are a reflective account common in social sciences research. In them, researchers acknowledge how their background, experiences, and biases may have influenced their work. If developers believe such factors significantly impacted their benchmark's construction, they may provide a positionality statement for increased context and transparency.

---

[5]Such statements were not included in the assessment to avoid pressuring benchmark developers to disclose potentially sensitive personal information, even if such information influenced the benchmark design process.

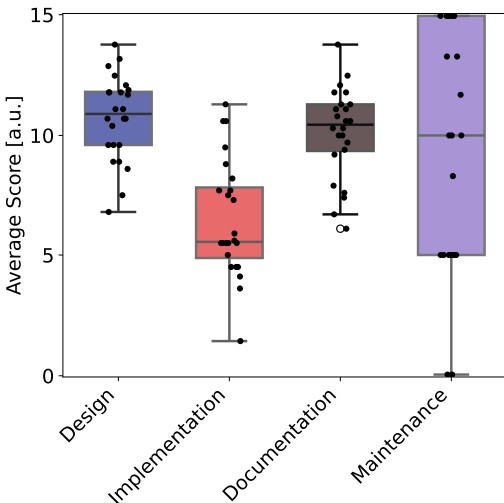

Figure 6: Average and individual scores of all assessed benchmarks per lifecycle stage.

| Stage | FM | Non-FM | All |
|---|---|---|---|
| **Design** | 10.6 | 11.1 | 10.7 |
| **Implementation** | 5.5 | 7.4 | 6.1 |
| **Documentation** | 10.3 | 9.9 | 10.1 |
| **Maintenance** | 9.1 | 10.8 | 9.7 |

Table 1: Benchmark lifecycle scores averaged over the 24 assessed benchmarks separated for FM, non-FM, and All benchmarks combined.

| | FM | Non-FM | All |
|---|---|---|---|
| **Pearson** $\rho$ | 0.721 | 0.318 | 0.655 |
| **p-value** $p$ | 0.001 | 0.487 | 0.001 |

Table 2: Pearson correlation coefficient for FM, Non-FM, and All benchmarks between the design and usability (weighted average of implementation, documentation, and maintenance stages) score as in Fig. 7.

## 6 Quantitative Results

In this section, we present our assessment results.[6] Tab. 1 showcases the average scores per benchmark lifecycle stage, showing that for both FM and non-FM benchmarks, the implementation stage tends to be the weakest area, followed by maintenance. All criteria averages are reported in App. F. Some criteria have not been fulfilled by almost any benchmark (e.g., *Standardized metadata is included*). Notably, both benchmark types are particularly weak for criteria supporting the reproducibility and interpretation of results: benchmarks get an average score of 3.75 on *Including a script to replicate results* and an average score of 5.62 on *Reporting statistical significance*.

While individual benchmark or criteria scores are deterministic, we can analyze statistical fluctuations across categories and benchmarks. Fig. 7 compares the design and usability scores of FM and non-FM benchmarks. The overall average design score across all benchmarks is 10.7, and the weighted average usability score is 8.7. The difference in mean design and usability scores between FM and non-FM benchmarks is not statistically significant (95% confidence level), see Fig. 8 in App. E. Furthermore, we find statistically significant correlations between the design and usability scores for FM benchmarks alone and all benchmarks combined at the 95% confidence level (Tab. 2). This suggests that, in both cases, benchmarks with poorer design tend to also be less usable, and vice versa.

## 7 Discussion

**Not all benchmarks are of the same quality.** Model developers frequently report performance on benchmarks that vary significantly in quality. For instance, the widely-used MMLU benchmark scored the lowest in our assessment (weighted average: 5.5), while GPQA scored significantly higher (weighted average: 11.0). However, recent communications introducing models like GPT-4 [3], Claude-3 [8], and Gemini [80] report results on both benchmarks without explicitly acknowledging their limitations or quality differences. This practice may be driven by the assumed expectation that reviewers want to see a wide range of metrics and the belief that readers should determine the most relevant metrics for their needs. The lack of clear guidance on AI benchmark quality and limitations may lead to incorrect conclusions about a model's performance, even if developers do not intend to

---

[6]Per-criterion scores for all benchmarks are released on our website betterbench.stanford.edu. Code to replicate results will be available on GitHub upon publication.

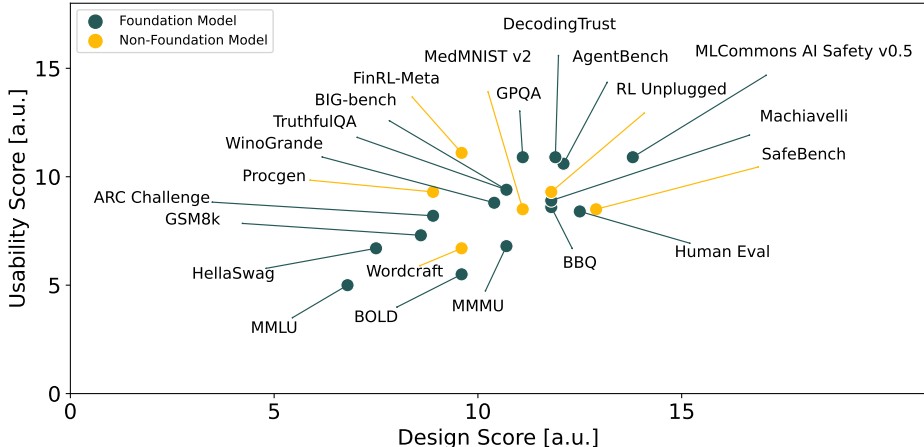

Figure 7: Design and usability score for all 24 assessed benchmarks. The usability score is the weighted average of the implementation, documentation, and maintenance scores. Benchmarks were split into foundation model and non-foundation model benchmarks, depending on the model group they're targeting.

mislead users. The UK AI Safety Institute's *Inspect* framework [81] similarly includes both MMLU [33] and GPQA [68], potentially resulting in misleading evaluations. This is problematic because governments increasingly rely on evaluations for AI regulations and may use frameworks like *Inspect* [69] or individual benchmarks [1].

**Most benchmarks fail to distinguish signal and noise.** Benchmark developers should not only report a single result for a model but also re-run their evaluation [13] with, e.g., different random seeds or sampling temperatures, and report the mean and variance for these intra-model evaluations. As benchmarks are primarily used to compare models, users must know the intra-model variance of a benchmark to determine whether observed inter-model variances are genuine performance differences or arise from noisy results. If intra-model variance bounds are tight and inter-model variance bounds are wide, benchmark users can conclude that there are genuine performance differences between models. However, if both intra- and inter-variance bounds are wide, statistical analysis is required to discern noise and actual signal. Yet, 14 out of 24 benchmarks did not perform multiple evaluations of the same model or report statistical significance or uncertainty of results.

**Insufficient implementation limits reproducibility and scrutiny of benchmarks.** Our analysis reveals that scores for implementation stage criteria are the lowest across all assessed benchmarks. Notably, 17 out of 24 benchmarks do not provide easy-to-run scripts to replicate the results reported in the initial paper, and 4 out of 24 only provide scripts to replicate part of the results. This lack of accessibility hinders reproducibility and limits users' ability to scrutinize the benchmarking process. In a field where reproducibility is a significant concern [43], providing materials to reproduce results is crucial for validating benchmark findings.

**Small changes can lead to significant improvements in overall benchmark practices.** Many of the criteria we have identified for improving AI benchmarks are relatively easy to implement, even for existing benchmarks. For example, adding code documentation and and a point of contact are not time consuming to add, yet can significantly enhance usability, accountability, and ease of use.

**Necessity for higher benchmark development standards.** As evidenced by the strong discrepancies in AI benchmark quality we found (Sec. 6 and App. F), there is a need to introduce additional checks for benchmarking practices to ensure a minimum quality standard for AI benchmarks. We assume that benchmark developers do not intentionally construct insufficient benchmarks, but rather do so due to limited knowledge of what constitutes a good benchmark. By providing a checklist of best practices (App. J.1), we aim to make it easy for benchmark developers to adopt these recommendations and

improve the quality of their benchmarks. In addition, some of the criteria we have identified in our expert interviews and from reviewing evaluation practices in other fields, such as including a build status in GitHub repositories that assesses whether the last commit successfully passed defined unit tests [28], were relatively unknown and only implemented by 3 out of 24 benchmarks. Other criteria, like using globally unique identifiers or encrypting evaluation instances to avoid data contamination, have been pioneered by only a few of the assessed benchmarks [68, 74] but have not yet gained widespread adoption. By incorporating these criteria into our assessment, we aim to encourage benchmark developers to adopt these best practices in the field of AI benchmarking.

## 8 Limitations

Our assessment assigns equal weight to all criteria, despite their varying levels of effort required for fulfillment and differing contributions to overall benchmark quality. The scoring system differentiates only four score categories to enable relatively objective evaluation through clear-cut criteria (App. K and App. G), but may miss nuances within each category. For example, a benchmark barely fulfilling a criterion and one almost entirely fulfilling it would receive the same 10-point score. Given the equal weighting and scoring, benchmark developers could potentially "game" the assessment by focusing on easily fulfilled criteria. However, we believe that even if a developer only implements easy-to-implement criteria, the resulting benchmark will still be of higher quality than one not meeting any criteria, thus fulfilling our work's goal. Furthermore, assessing the construct validity of a benchmark and determining whether its approach to assessing a concept is truly effective would presumably require in-depth analysis by domain experts in the respective fields, which is beyond the scope of this assessment. Instead, we aim to provide benchmark developers with a blueprint for minimum quality assurances. Finally, our framework is intended for public benchmarks and future work is needed to extend it to private ones.

## 9 Impact Statement

By releasing the first systematic assessment framework for AI benchmarks, we aim to encourage benchmark developers to construct higher-quality benchmarks and to contribute to community efforts to make AI evaluations more practicable and transparent. Higher-quality benchmarks resulting from the adoption of our framework and checklist can lead to better-informed model selection for downstream tasks, potentially reducing risks and improving outcomes in high-stakes applications. Our living repository of benchmark assessments promotes transparency and comparability, allowing benchmark users to make informed decisions when choosing benchmarks. However, there is a potential risk of misinterpretation of our results; our assessment only provides minimum quality assurances and is not sufficient to assess the suitability of a benchmark for a concrete use case. The outputs of our evaluation do not contain sensitive or harmful content, but users may encounter such content during a benchmark assessment depending on the benchmark's data. While we do not anticipate direct safety risks from releasing our framework, we acknowledge that strict adherence to some of our proposed criteria, such as the involvement of domain experts, may unequally impact researchers based on their access to resources and connections, potentially hindering the development of benchmarks from a broader range of research institutions and underrepresented communities, which could limit diversity in benchmark creation.

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

# A   Open Challenges in AI Benchmarking

Per the current state of the field, some benchmark issues are not fully addressable by benchmark developer actions and decisions. This section discusses these issues and directs readers, where possible, to resources which cover these open problems in greater depth.

**Quick saturation.** Rapid advancements in AI have led to the saturation of many benchmarks. Some benchmarks have been saturated within months of their release [58]. Addressing this issue involves evaluating current model performances and assessing whether the concept has already been solved, and determining if the benchmark can be made challenging given state-of-the-art capabilities of the models tested.

**Contamination.** In Sec. 4.2, we discuss strategies to mitigate data contamination. However, even when fully adhered to, challenges remain. For example, benchmark developers cannot enforce model developers' use of canary strings to avoid training on benchmark data. Preventing data contamination, particularly in models reliant on large amounts of web-scraped data, is a shared responsibility between benchmark and model developers. [90] offers further description of measures that can be taken on the model developer side. This issue is pressing, as contamination has been demonstrated in both FM [29, 37, 47] and non-FM [43, 41]. Future work across stakeholders is needed to effectively mitigate contamination and preserve benchmark validity.

**Poor construct validity.** Construct validity refers to the degree to which a test or measurement tool accurately measures the construct it intends to measure [22]. [61] outline factors which make construct validity, especially in FM benchmarking, a challenge. They describe certain properties (e.g. factual accuracy) that arise from the interaction between the model and its user population, rather than from the model alone. To combat this, they suggest incorporating ecologically valid[7] user interactions into the assessment; yet, given the lack of transparency by model developers into actual user interactions, this criteria is difficult to implement for benchmark developers. Alternately, [23] propose that guarantees be made through formal verification, although this approach has not yet been tested in practice.

**Standardization of benchmark reporting.** Due to the difficulties with construct validity, most benchmarks cannot provide an absolute signal and instead give a relative one by comparison of models on the same benchmark. This signal is often unavailable to potential model users, as there is no present standardization of benchmark reporting. Model developers report whichever benchmarks they see fit without being obligated to provide a rationale, resulting in inconsistent reporting, especially apparent in the case of benchmarks relating to responsible AI concepts [58]. While this issue does not depend on further research, there is no consensus in theory or practice regarding how benchmark reporting should be standardized. Potential avenues towards standardization include publication of benchmark results through independent entities, market incentives such as government contracts, and mandatory reporting as part of AI legislation.

# B   Stakeholders

This section details the stakeholders that are involved in benchmark development and use processes.

**Benchmark developers.**   Benchmark developers are the individuals or teams who create benchmarks from scratch (e.g. BIG-Bench [74]), by expanding on previously developed benchmarks (e.g. MedMNIST v2 [89]), by integrating multiple existing benchmarks (e.g. HELM [48]), or by both expanding upon and integrating other benchmarks (e.g. Decoding Trust [84]). This groups objectives are developing benchmarks that accurately and comprehensively assess models capabilities or safety-critical characteristics and establishing standards for AI system evaluations that facilitate comparisons and drive progress on the specified tasks. There are three use cases for benchmark developers of our assessment, checklist, and website:

---

[7]Ecological validity is the extent to which the findings of a research study are able to be generalized to real-life settings [46]

- They use the checklist to understand best practices and guide their benchmark construction process pre-deployment.

- They use the assessment to score their benchmark after constructing it to understand any shortcomings they may address to improve the overall benchmark quality.

- They can use the website to find related benchmarks and compare their benchmark quality to those.

**Model developers.**   Model developers are the individuals or teams who develop AI models for commercial use (e.g. GPT-4 [3]) or non-commercial purposes (e.g. Alpaca [79]). Their objectives in using benchmarks are demonstrating the performance of their models identifying areas for improvement which can guide model development and to establish credibility and encourage adoption by showcasing favorable relative performance. There are three use case for model developers of our assessment and website:

- They can use the assessment results to decide which benchmarks to report

- Model developers can reference our assessment results in their official reporting to indicate quality differences between benchmarks, if applicable

- Model developers can use our website to find relevant benchmarks to report for their model

**Model users.**   Model users are the individuals, organizations, or businesses which use or modify available AI models for various downstream applications (e.g. a company using ChatGPT to provide customer service). Their objective when using benchmark results is making informed decisions regarding which AI models are most suitable for their specific use cases. There are two use case for model users of our assessment and website:

- If model developers dont reference our or any similar benchmark quality assessment, model users can refer to our assessment results on the website to understand quality differences in benchmarks reported by model developers.

- They can also refer to our benchmark assessment results to decide between two related benchmarks who's results may both be relevant for the model user's application context. If one of these benchmarks has a higher quality, they may decide to prioritize that result based on our assessment.

**AI researchers.**   AI researchers are individuals or teams studying AI and related fields either at non-profits, within academic institutions, in industry, or independently. One of researchers objectives is using benchmarks to evaluate the performance of novel AI architectures, training techniques, and approaches, and to compare these to other systems. Additionally, they have the objective of setting research agendas based on the model limitations and weaknesses revealed by benchmarks. There are two use case for AI researchers of our assessment and website:

- Based on our website and assessment results, AI researchers may analyze benchmarking practices in more detail to understand challenges of benchmark developers and drive research on open questions in AI evaluations and AI benchmarking more broadly.

- They can use our website to understand the overall AI benchmark landscape.

**Regulators and standard-setting organizations.**   Regulators and standard-setting organizations may be affiliated with government agencies, international bodies, and industry associations. In these roles, they are responsible for creating and enforcing standards and regulations for AI development and use. Examples of such entities are the AI Safety Institutes, the ISO, and the EU Commission. The objective of these stakeholders is using benchmarks to assess the compliance of AI models with established regulations, guidelines and standards for traits such as performance, fairness, and safety. For example, the UK AI Safety Institute recently released their *Inspect* evaluation framework [81] that includes several benchmarks that we scored in our assessment, among other evaluation strategies. There are two use case for model users of our assessment and website:

- Regulators and standard-setting organizations can refer to our checklist to design regulatory requirements, e.g., by only accepting benchmarks as proof for compliance by model developers that completed certain or all criteria in our checklist

- They can also mandate that only benchmarks that achieved a certain score on our assessment may be used to proof compliance with regulatory requirements.

## C  Benchmark Lifecycle

**Design.**  During the design stage, a benchmarks purpose, scope, and structure are defined. This requires developers to identify key aspects of an AI system that the benchmark will assess. Based on this decision, they must determine the tasks, datasets, and evaluation metrics which will be used in their benchmark. To inform these decisions, developers consider the requirements of potential users, possibly collaborating with and gathering feedback from these and other stakeholders.

**Implementation.**  At this stage, the benchmark is constructed and all necessary components are aggregated. Developers collect, process, and (if applicable) annotate the datasets to be used for their tasks. They then create the evaluation scripts which allow models performance on this data to be measured. So that new models can be evaluated, developers may implement user interfaces and APIs which enable access to and interaction with the benchmark. This stage concludes with the initial testing and validation of benchmark components.

**Documentation.**  To facilitate the benchmarks use and interpretation, benchmark developers need to create comprehensive documentation. This includes preparing detailed descriptions of benchmark tasks, datasets, and evaluation metrics. Additionally, developers may provide instructions for how to access, use, and submit to the benchmark. Documenting design decisions, limitations, and potential biases enables stakeholders to make informed decisions regarding benchmark use. Creating resources for running the benchmark, such as quick-start guides, code documentation, and examples or tutorials is an essential step for accessibility.

**Maintenance.**  Once the benchmark and its documentation are released, developers must conduct regular maintenance to ensure ongoing usability. They may monitor benchmark usage and performance to identify areas for improvement and track users compliance with release requirements. Other tasks at this stage include addressing issues or bugs and incorporating user feedback into updates. Developers can regularly update documentation and support materials. Additionally, they can assess the continued relevance and utility of the benchmark by monitoring performance on the benchmark and responding to community feedback.

**Retirement.**  The final phase of a benchmarks lifecycle is retirement. Benchmarks are phased out or replaced when they become saturated (i.e. model performance reaches the benchmark metrics ceiling), the task studied loses relevance, or better alternatives emerge. During retirement, developers communicate their plan to stakeholders and can provide guidance on transitioning to alternatives. They archive benchmark data, code, and documentation. As a benchmark is retired, developers may share insights gained with the AI community. Finally, they should clearly mark the benchmark as "retired" on channels for deployment and platforms publishing its results.

## D  List of Assessed Benchmakrs

We evaluate these 16 foundation model benchmarks (alphabetical order):

- AgentBench [51]
- ARC Challenge [19]
- BBQ [64]

- BIG-bench [74]
- BOLD [26]
- Codex HumanEval [17]
- DecodingTrust [84]
- GPQA [68]
- GSM8k [21]
- HellaSwag [93]
- Machiavelli [63]
- MLCommons AI Safety v0.5 [82]
- MMLU [33]
- MMMU [92]
- TruthfulQA [50]
- WinoGrande [71]

We evaluate these 8 non-foundation model benchmarks (alphabetical order):

- ALE [11]
- FinRL-Meta [53]
- MedMNIST v2 [89]
- PDEBench [78]
- Procgen [20]
- RL Unplugged [31]
- SafeBench [88]
- Wordcraft [38]

# E   Sensitivity Analysis Details

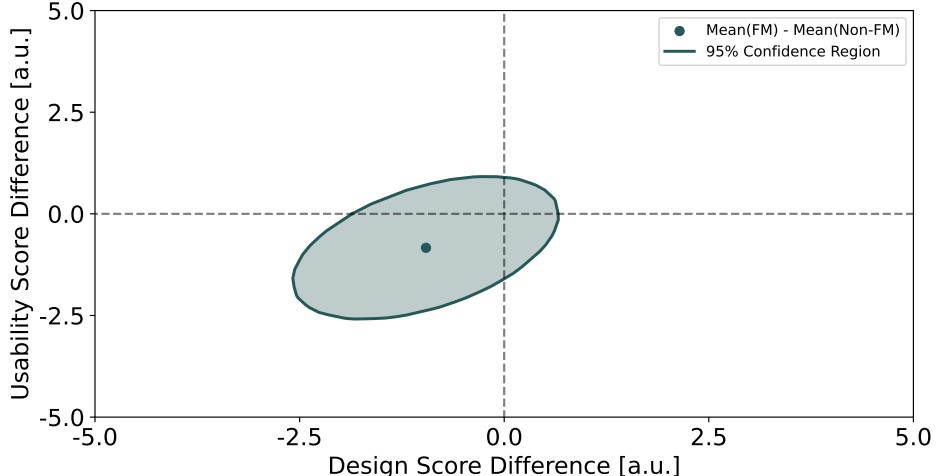

Figure 8: Calculating the difference between the mean Usability and Design score between foundation model (FM) and non-foundation model (Non-FM) benchmarks with the data in Fig. 8. We show the lack of statistical significance of the difference using bootstrap resampling at a 95% confidence level.

We show that the difference in mean usability score between FM and non-FM benchmarks in Fig. 8 is not statistically significant using bootstrap resampling at a 95% confidence level.

## F  Additional Results

All individual benchmark scoring results, including justifications, can be found on *betterbench.stanford.edu*.

### F.1  Scores per lifecycle Stage

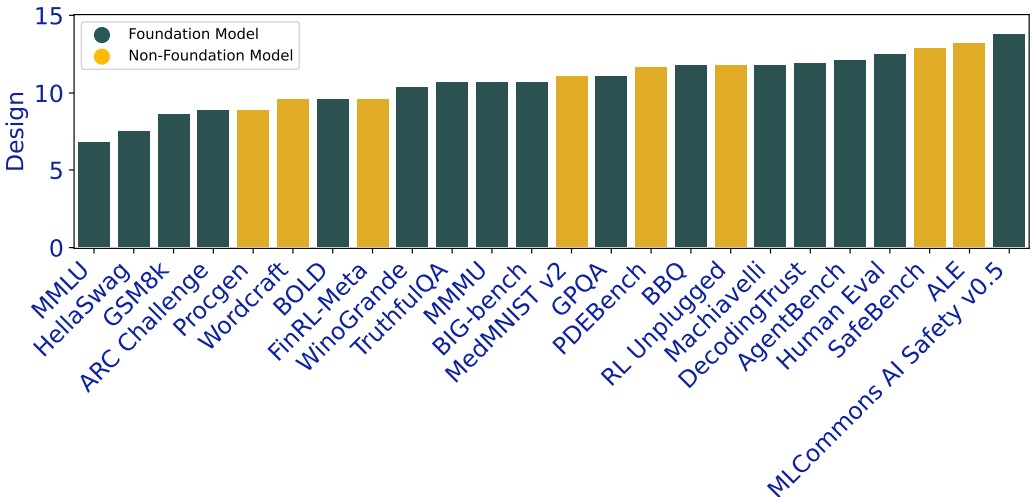

Figure 9: In ascending order, design scores for each benchmark, separated for foundation model (FM) and non-foundation model (Non-FM) benchmarks.

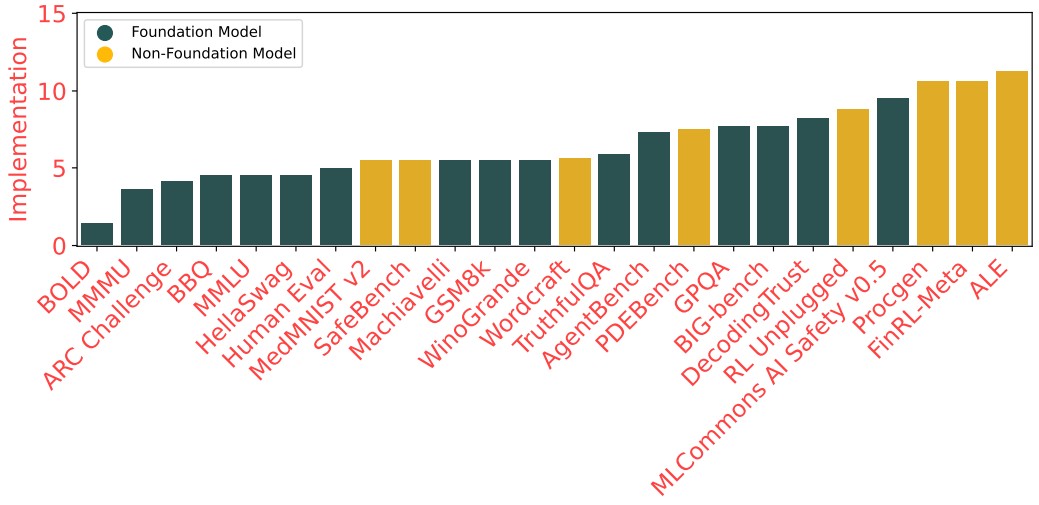

Figure 10: In ascending order, implementation scores for each benchmark, separated for foundation model (FM) and non-foundation model (Non-FM) benchmarks.

We show the scores for each benchmark and for each benchmark lifecycle stage as barplots (Design: Fig. 9, implementation: Fig. 10, documentation: Fig. 11, and maintenance Fig. 12). The scores for each benchmark for each individual category can be found on our website, betterbench.stanford.edu. For the bar plots for each stage, the benchmarks are shown in ascending order and marked as FM and non-FM benchmark.

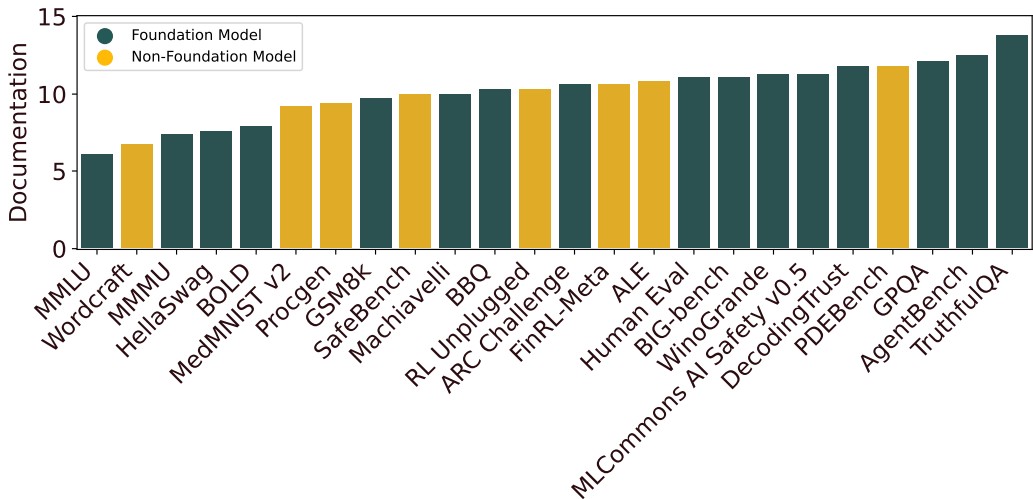

Figure 11: In ascending order, documentation scores for each benchmark, separated for foundation model (FM) and non-foundation model (Non-FM) benchmarks.

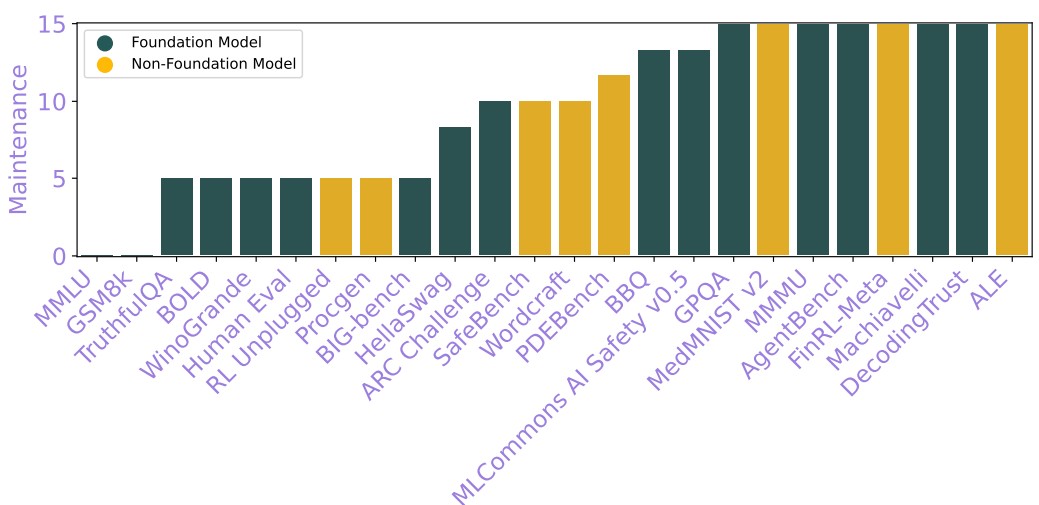

Figure 12: In ascending order, maintenance scores for each benchmark, separated for foundation model (FM) and non-foundation model (Non-FM) benchmarks.

# G  Scoring

We evaluate 24 benchmarks based on criteria grouped into category (a) (see Sec. 3), i.e., those controlled by the benchmark developer where the authors and interviewees reached a normative consensus. We use the following discrete point system to score each criteria:

- Criteria not acknowledged and not addressed: 0 points

- Criteria acknowledged but not addressed: 5 points

- Criteria partially addressed: 10 points

- Criteria fully addressed: 15 points

- Criteria not relevant: n/a

The highest possible score per category is 15, and the lowest is 0. The criteria span the benchmark lifecycle stages of design, implementation, documentation, and maintenance. Benchmark retirement is excluded from the assessment and scoring, since most benchmarks we looked at are still actively used and not saturated, and given that we cannot predict/anticipate if benchmark developers would in fact fulfill any criteria we'd list for this category. All individual evaluations are made publicly available.

For each lifecycle stage, we calculate the average points earned across the relevant criteria for that stage, excluding any criteria scored as "n/a". This results in four subscores:

- $s_D$ = Design score
- $s_I$ = Implementation score
- $s_{Do}$ = Documentation score
- $s_M$ = Maintenance score

We do not differentiate the importance of criteria or effort to address them within each lifecycle stage, weighting them equally in the average. To provide an overall assessment of a benchmark's design and usability, we aggregate the subscores into two key measures:

- Design score $S_D$:
  - Showcases how clear about a benchmark is about its intended purpose and scope and how interpretable it is
  - Equivalent to the design stage subscore $s_D$
- Usability score $S_U$:
  - Indicates how easy the benchmark is use and how well it is documented and maintained
  - Weighted average of the implementation, documentation, and maintenance scores, see Equ. 1.

$$S_U = \frac{n_I s_I + n_{Do} s_{Do} + n_M s_M}{n_I + n_{Do} + n_M} \tag{1}$$

Where:

- $S_U$ represents the usability score
- $s_I$ represents the implementation score
- $s_{Do}$ represents the documentation score
- $s_M$ represents the maintenance score
- $n_I$ represents the number of criteria in the implementation stage that are not n/a for the respective benchmark
- $n_{Do}$ represents the number of criteria in the documentation stage that are not n/a for the respective benchmark
- $n_M$ represents the number of criteria in the maintenance stage that are not n/a for the respective benchmark

The discrete 0/5/10/15 point scale provides clearer differentiation between criteria that are not addressed, partially addressed, and fully addressed compared to a continuous scale. At the same time, it allows for a quantitative analysis compared to a letter grade scale like A/B/C/D. Allowing for an N/A option handles criteria that may not be applicable to certain benchmarks. The 0/5/10/15 scale also allows for more granular distinctions compared to a narrower scale like 0/1/2/3 in the final scores: The difference between a score of 5 (acknowledged but not addressed) and 10 (partially addressed) is easier to see than between a 2 and 3 on a narrower scale. With a smaller range, the difference between scores is less meaningful and it is harder to separate the varying degrees of benchmark quality. Providing subscores for each lifecycle stage, while rolling them up into overall Design and Usability Scores, enables assessing benchmarks at both a category and aggregate level.

## H Methodology Flow Diagram

Fig. 13 shows a detailed overview of the steps we took to derive the best practices that formed the basis of our AI benchmark assessment.

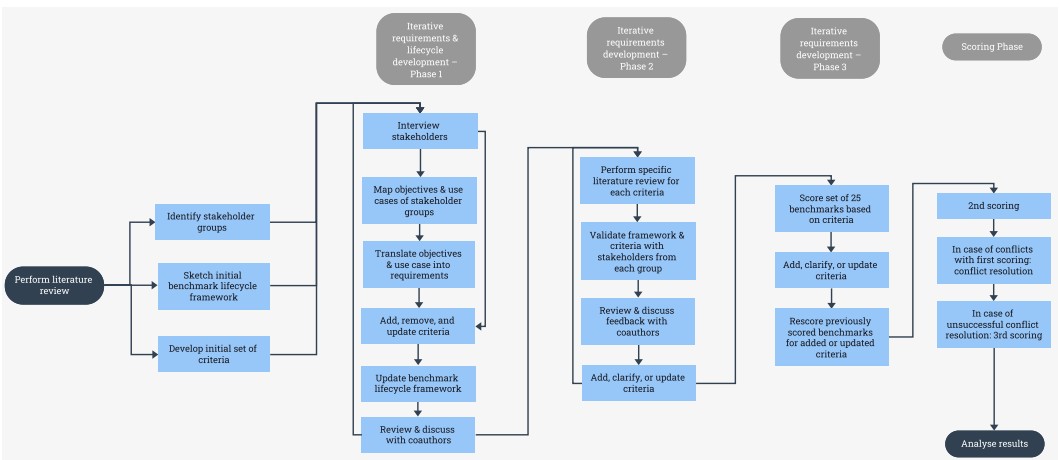

Figure 13: Flow diagram showing our detailed process how we derived the best practices for benchmarks.

## I Release Requirements

1. Benchmark developers acknowledge that our checklist is a minimum quality assurance and not sufficient for high-quality benchmark construction.

2. Benchmark developers do not attempt to game our assessment, e.g. by just changing the code checked update on the GitHub repository side without actually checking their code's usability.

## J BetterBench Checklist for Benchmark Developers

In this section, we provide the assessment criteria as a checklist for benchmark developers to use during their benchmark construction process, pre-deployment of the benchmark. If benchmark developers want to list their benchmark on our website, they will also have to submit this checklist. On the website, we will further provide an easy-to-fill-out checklist in LATEXand .doc format that can be easily included as part of any benchmark documentation. In the second subsection, we will also add an example of a filled out checklist assessing BetterBench, which can be seen as a benchmark for benchmarks. Going through the checklist was part of the validation of our methodology, described in Step 4 of the Sec. 3 section.

### J.1 Template

- **Benchmark Design**

  ☐ The tested capability, characteristic, or concept is defined
  – TODO | YES | NO | N/A
  – Justification:

  ☐ How tested capability or concept translates to benchmark task is described
  – YES | NO | N/A
  – Justification:

  ☐ How knowing about the tested concept is helpful in the real world is described.

1011     – YES | NO | N/A

1012     – Justification:

1013 ☐ How benchmark score should or shouldn't be interpreted/used is described

1014     – YES | NO | N/A

1015     – Justification:

1016 ☐ Domain experts are involved

1017     – YES | NO | N/A

1018     – Justification:

1019 ☐ Use cases and/or user personas are described

1020     – YES | NO | N/A

1021     – Justification:

1022 ☐ Domain literature is integrated

1023     – YES | NO | N/A

1024     – Justification:

1025 ☐ Informed performance metric choice

1026     – YES | NO | N/A

1027     – Justification:

1028 ☐ Metric floors and ceilings are included

1029     – YES | NO | N/A

1030     – Justification:

1031 ☐ Human performance level is included

1032     – YES | NO | N/A

1033     – Justification:

1034 ☐ Random performance level is included

1035     – YES | NO | N/A

1036     – Justification:

1037 ☐ Automatic evaluation is possible and validated

1038     – YES | NO | N/A

1039     – Justification:

1040 ☐ Differences to related benchmarks are explained

1041     – YES | NO | N/A

1042     – Justification:

1043 ☐ Input sensitivity is addressed

1044     – YES | NO | N/A

1045     – Justification:

1046 • **Benchmark Implementation**

1047 ☐ The evaluation code is available

1048     – YES | NO | N/A

1049     – Justification:

1050 ☐ The evaluation data or generation mechanism is accessible

1051     – YES | NO | N/A

1052     – Justification:

1053 ☐ The evaluation of models via API is supported

1054     – YES | NO | N/A

1055     – Justification:

1056 ☐ The evaluation of local models is supported

1057     – YES | NO | N/A

– Justification:

☐ A globally unique identifier is added or evaluation instances are encrypted

  – YES | NO | N/A

  – Justification:

☐ A task to identify if model is included trained on benchmark data

  – YES | NO | N/A

  – Justification:

☐ A script to replicate results is explicitly included

  – YES | NO | N/A

  – Justification:

☐ Statistical significance or uncertainty quantification of benchmark results is reported

  – YES | NO | N/A

  – Justification:

☐ Need for warnings for sensitive/harmful content is assessed

  – YES | NO | N/A

  – Justification:

☐ A build status (or equivalent) is implemented

  – YES | NO | N/A

  – Justification:

☐ Release requirements are specified

  – YES | NO | N/A

  – Justification:

- **Benchmark Documentation**

☐ Requirements file or equivalent is available

  – YES | NO | N/A

  – Justification:

☐ Quick-start guide or demo is available

  – YES | NO | N/A

  – Justification:

☐ In-line code comments are used

  – YES | NO | N/A

  – Justification:

☐ Code documentation is available

  – YES | NO | N/A

  – Justification:

☐ Accompanying paper is accepted at peer-reviewed venue

  – YES | NO | N/A

  – Justification:

☐ Benchmark construction process is documented

  – YES | NO | N/A

  – Justification:

☐ Test tasks & rationale are documented

  – YES | NO | N/A

  – Justification:

☐ Assumptions of normative properties are documented

  – YES | NO | N/A

  – Justification:

☐ Limitations are documented
  – YES | NO | N/A
  – Justification:

☐ Data collection, test environment design, or prompt design process is documented
  – YES | NO | N/A
  – Justification:

☐ Evaluation metric is documented
  – YES | NO | N/A
  – Justification:

☐ Applicable license is specified
  – YES | NO | N/A
  – Justification:

- **Benchmark Maintenance**

  ☐ Code usability was checked within the last year
    – YES | NO | N/A
    – Justification:

  ☐ Maintained feedback channel for users is available
    – YES | NO | N/A
    – Justification:

  ☐ Contact person is listed
    – YES | NO | N/A
    – Justification:

## J.2 Example

As noted in Sec. 3, we assessed BetterBench against our own assessment framework to verify that the framework is usable and practiable. This section showcases this assessment and gives an example of a filled-out checklist, based on the template provided in App. J.1,

- **Benchmark Design**

  ☐ The tested capability, characteristic, or concept is defined
    – YES
    – Justification: "We define a *high-quality* benchmark to be one that is clear about its intended purpose and scope, and that is usable. To date, no structured assessment for the quality of AI benchmarks, including both FM and non-FM benchmarks, has been published to date, and no comparative analysis was conducted to understand quality differences between widely used benchmarks in the field. This paper addresses these gaps"(Sec. 1)

  ☐ How tested capability or concept translates to benchmark task is described
    – YES
    – Justification: For detail, see Sec. 4 and App. K

  ☐ How knowing about the tested concept is helpful in the real world is described.
    – YES
    – Justification: Justification: "By releasing the first systematic assessment framework for AI benchmarks, we aim to encourage benchmark developers to construct higher-quality benchmarks and to contribute to community efforts to make AI evaluations more practicable and transparent. Higher-quality benchmarks resulting from the adoption of our framework and checklist can lead to better-informed model selection for downstream tasks, potentially reducing risks and improving outcomes in high-stakes applications" (Sec. 9).

- [ ] How benchmark score should or shouldn't be interpreted/used is described
  - YES
  - Justification: "Our living repository of benchmark assessments promotes transparency and comparability, allowing benchmark users to make informed decisions when choosing benchmarks. However, there is a potential risk of misinterpretation of our results; our assessment only provides minimum quality assurances and is not sufficient to assess the suitability of a benchmark for a concrete use case" (Sec. 9).
- [ ] Domain experts are involved
  - YES
  - Justification: "Initially, we surveyed the existing benchmark landscape (Sec. 2). Based on this review, we identified five stakeholder groups who present the user personas of our assessment (App. B). All stakeholder groups were represented in subsequent unstructured interviews which included 20+ policymakers, model developers, benchmark developers, model users, and AI researchers, to understand their objectives w.r.t. benchmarking. During this process, we developed a five-stage model of the benchmark lifecycle (Fig. 5 and App. C) and mapped the benchmarking objectives of the stakeholders, along with their communicated use cases of a benchmark assessment (App. B)" (Sec. 3).
- [ ] Use cases and/or user personas are described
  - YES
  - Justification: "We identified five stakeholder groups who present the user personas of our assessment" (Sec. 3, see full personas and use cases in App. B).
- [ ] Domain literature is integrated
  - YES
  - Justification: "Our work is informed by benchmarking practices from fields beyond AI, ranging from transistor hardware [18] to environmental quality [16] to bioinformatics [7], and identify common themes regarding what constitutes an effective benchmark. When applicable, we incorporate these best practices into our assessment (Sec. 4)." Citations for this literature, when used, are provided in Sec. 4.
- [ ] Informed performance metric choice
  - YES
  - Justification: "The discrete 0/5/10/15 point scale provides clearer differentiation between criteria that are not addressed, partially addressed, and fully addressed compared to a continuous scale. At the same time, it allows for a quantitative analysis compared to a letter grade scale like A/B/C/D. Allowing for an N/A option handles criteria that may not be applicable to certain benchmarks." Full details on our scoring method are available in App. G.
- [ ] Metric floors and ceilings are included
  - YES
  - Justification: "The highest possible score per category is 15, and the lowest is 0" (App. G).
- [ ] Human performance level is included
  - N/A
  - Justification: In our work, we manually evaluate AI benchmarks; a human could not be used as an evaluation target in our context.
- [ ] Random performance level is included
  - N/A
  - Justification: Random generation cannot constitute an AI benchmark.
- [ ] Automatic evaluation is possible and validated
  - N/A

   – Justification: "Given the varying information sources (official websites, papers, GitHub repositories published by the benchmark developers that we do consult to assess benchmarks, and given that they do not follow a standard structure, we manually evaluate all benchmarks" (Sec. 3).

  ☐ Differences to related benchmarks are explained

   – YES

   – Justification: "Unlike prior studies, such as [59] and [49], which focus on identifying the limitations, our approach offers a practical evaluation, empowering developers to address shortcomings and enhance benchmark quality directly" (Sec. 2.1).

  ☐ Input sensitivity is addressed

   – N/A

   – Justification: Since our benchmark uses human evaluation, we select a single phrasing for each criterion. As described in Sec. 3 these phrasings were developed iteratively to maximize clarity and minimize disagreement amongst multiple annotators of the same benchmmark.

- **Benchmark Implementation**

  ☐ The evaluation code is available

   – N/A

   – Justification: We performed human evaluation which did not use code.

  ☐ The evaluation data or generation mechanism is accessible

   – N/A

   – Justification: We evaluate benchmarks based on "official websites, papers, GitHub repositories published by the benchmark developers" (Sec. 3). The availability of these materials is dependent on benchmark developers.

  ☐ The evaluation of models via API is supported

   – N/A

   – Justification: We evaluate benchmarks rather than models.

  ☐ The evaluation of local models is supported

   – N/A

   – Justification: We evaluate benchmarks rather than models.

  ☐ A globally unique identifier is added or evaluation instances are encrypted

   – N/A

   – Justification: Our benchmark does not evaluate AI models or include any examples which they could be contaminated by training on.

  ☐ A task to identify if model is included trained on benchmark data

   – N/A

   – Justification: Our benchmark does not evaluate AI models or include any examples which they could be contaminated by training on.

  ☐ A script to replicate results is explicitly included

   – N/A

   – Justification: The code to replicate results will be added as supplementary material and published as part of a GitHub repo upon publication.

  ☐ Statistical significance or uncertainty quantification of benchmark results is reported

   – YES

   – Justification: These results are reported in Sec. 6 and App. E.

  ☐ Need for warnings for sensitive/harmful content is assessed

   – YES

   – Justification: "The outputs of our evaluation do not contain sensitive or harmful content, but users may encounter such content during a benchmark assessment depending on the benchmark's data" (Sec. 9).

- A build status (or equivalent) is implemented
  - YES
  - Justification: A build status will be included in the code released as part of a GitHub repo upon publication.
- Release requirements are specified
  - YES
  - Justification: Release requirements are provided in App. I.

- **Benchmark Documentation**

  - Requirements file or equivalent is available
    - YES
    - Justification: A requirements file will be included in the code released as part of a GitHub repo upon publication.
  - Quick-start guide or demo is available
    - YES
    - Justification: We provide a checklist to facilitate use of our benchmark in App. J and an example of its use in App. J.2. Additionally, we will include a quick-start guide for our code in the GitHub repo released upon publication.
  - In-line code comments are used
    - YES
    - Justification: Our GitHub repository includes in-line code comments.
  - Code documentation is available
    - YES
    - Justification: Our GitHub repository includes code documentation.
  - Accompanying paper is accepted at peer-reviewed venue
    - N/A
    - Justification: Our paper is currently under submission at a peer-reviewed venue.
  - Benchmark construction process is documented
    - YES
    - Justification: We describe our full process in Sec. 3.
  - Test tasks & rationale are documented
    - YES
    - Justification: Definitions and justifications for all criteria are presented in App. K.
  - Assumptions of normative properties are documented
    - YES
    - Justification:
  - Limitations are documented
    - YES
    - Justification: We discuss limitations in Sec. 8.
  - Data collection, test environment design, or prompt design process is documented
    - YES
    - Justification: We describe how we performed our evaluations in Sec. 3.
  - Evaluation metric is documented
    - YES
    - Justification: "We define a *high-quality* benchmark to be one that is interpretable and clear about its intended purpose and scope, and that is usable" Sec. 1. We further describe how we operationalized "quality" and calculate its subcomponents (design and usability) in Fig. 9 and Sec. 3.
  - Applicable license is specified

- YES
- Justification: We release our assessment under CC BY 4.0 license, available on our website (Sec. 3).

- **Benchmark Maintenance**

    ☐ Code usability was checked within the last year
    - YES
    - Justification: We have checked the usability of the code in our GitHub repository and will verify it again upon publication.

    ☐ Maintained feedback channel for users is available
    - YES
    - Justification: "Finally, we develop a supplementary website to continuously publish assessment results using the scoring methodology in App. G, given the rapid development of new benchmarks. The website includes a community feedback channel for submitting new AI benchmarks and correcting previously posted scores if benchmarks are updated or stakeholders disagree with our evaluation" (Sec. 3).

    ☐ Contact person is listed
    - YES
    - Justification: Contact details will be listed on our website.

# K Full Assessment Criteria

## K.1 Benchmark Design

1. **Definition of tested capability or characteristic**

    - **Explanation:** The benchmark developers mention and define what underlying capability or characteristic of a model is supposed to be tested with the benchmark.
    - **Justification:** Defining the objective of the benchmark is necessary for clarity in its design. It also helps users determine if the benchmark aligns with their specific application needs and ensures that users and developers have a shared understanding of the concept being evaluated, facilitating consistent interpretation of results.
    - **Points:**
        - 0: Tested concept, capability, or characteristic not explicitly mentioned.
        - 5: Tested concept explicitly mentioned and need for definition acknowledged, but definition not provided.
        - 10: Tested concept, capability, or characteristic explicitly mentioned but not defined.
        - 15: Tested concept, capability, or characteristic explicitly mentioned and defined.

2. **Description of how tested capability or concept translates to benchmark task**

    - **Explanation:** The benchmark developers describe how the tested capability or characteristic translates to the task implemented in the benchmark/the task the model is tested on in the benchmark.
    - **Justification:** Clearly explaining this translation ensures that the benchmark tasks accurately reflect the intended tested capabilities and concepts, providing valid assessment results.
    - **Points:**
        - 0: No description of how the tested capability or concept translates to the benchmark task.
        - 5: Acknowledgement that not describing how the tested capability or concept translates to the benchmark task is an issue, but no description provided.
        - 10: Description of how tested capability or concept translates to benchmark tasks provided for some but not all tasks.

       – 15: Description of how tested capability or concept translates to benchmark tasks provided for all tasks.

3. **Description of how knowing about the tested concept is helpful in the real world**

    • **Explanation:** The developers describe why it is useful to know about the tested capability in the real world.

    • **Justification:** This description helps users understand the practical value of the benchmark, demonstrating how the tested capability impacts real-world applications and use cases.

    • **Points:**

       – 0: No description of how knowing about the tested concept is helpful in the real world.

       – 5: Acknowledgement that not describing how knowing about the tested concept is helpful in the real world is an issue, but no description provided.

       – 10: Limited description of how knowing about the tested concept is helpful in the real world.

       – 15: Full description of how knowing about the tested concept is helpful in the real world.

4. **Description of use cases and user personas for the benchmark**

    • **Explanation:** A use case for an AI benchmark involves specifying a scenario in which the AI system will be evaluated. This scenario should include the cultural and geographic context and the type of interactions between humans and models [82], if applicable. Additionally, user personas should be defined to represent the different types of users that might interact with the AI system, if applicable. As a concrete example, [82] states "The use case for the v0.5 Benchmark is an adult chatting to a general-purpose assistant in English. The cultural and geographic context is Western Europe & North America. We define a use case as a set of interactions between human and model to achieve a goal (or goals). [...] For the v0.5 Benchmark, we are focusing on three personas: (i) a typical adult user; (ii) an adult user intent on malicious activities, behaving in a technically non-sophisticated way; and (iii) an adult user at risk of harm, behaving in a technically non-sophisticated way."

    • **Justification:** Use cases set the context and scope of the benchmark. User personas outline an understanding of the different types of interactions the benchmark developers anticipate the tested AI system to be used in, e.g., ranging from typical users to those with specific challenges or malicious intent. This approach ensures that the design of the benchmark is closely related to real-world applications and that it's effective across diverse scenarios.

    • **Points:**

       – 0: The benchmark does not include any description of use cases or user personas.

       – 5: The benchmark acknowledges the importance of use cases or user personas but does not explicitly formulate or describe them.

       – 10: The benchmark provides a partial description of use cases or user personas.

       – 15: The benchmark fully describes use cases and user personas, specifying the cultural and geographic context, types of human-model interactions (if applicable), and representing different user types that might interact with the AI system (if applicable).

       – n/a: For AI systems that do not involve direct human interaction, such as those used in industrial automation or scientific simulations, defining user personas is not relevant. However, real-world use cases should still be specified; in more theoretical benchmarks, this use case might be to advance research.

5. **Involvement of domain experts**

- **Explanation:** Domain expert(s) who have a professional background or research experience in the concept to be tested are either co-authors of the paper, or were involved in the benchmark design process, i.e., the paper makes clear how they obtained the expertise and how that informed the benchmark design.

- **Justification:** Involving domain experts ensures that the benchmark design is informed by deep, specialized knowledge, increasing its validity and relevance. This expertise helps to create tasks that accurately assess the targeted capabilities and align with real-world scenarios.

- **Points:**
  - 0: None of the authors has a background in the benchmark domain and no external experts were consulted during the design process.
  - 5: The benchmark mentions the necessity for in-domain expertise but doesn't specify any further details.
  - 10: The benchmark mentions that domain experts were consulted but not how their insights influenced the benchmark design.
  - 15: At least one of the co-authors has a professional or academic background in the benchmark domain or the benchmark specified how external experts were consulted and how that influenced the design process.

6. **Integration of domain literature**

- **Explanation:** The developers cite domain literature in the background section and describe how insights from this literature informed the design of their benchmark or cite relevant domain literature in the benchmark design process.

- **Justification:** By consulting domain-specific literature, benchmark developers can ensure that the tasks and evaluation criteria they include are representative and aligned with the current state of knowledge in the field. This literature often contains valuable insights into best practices, established methodologies, and proven approaches for evaluating the tested concept, which can be incorporated into the benchmark design to enhance its reliability.

- **Points:**
  - 0: The benchmark does not reference domain-specific literature.
  - 5: The benchmark mentions the need to integrate domain literature but did not address it in the background section or design process.
  - 10: The benchmark references domain literature in the background or related work section but does not describe how that domain literature informed the benchmark design process.
  - 15: The benchmark references domain literature throughout the paper and describes how that domain literature informed the benchmark design process.

7. **Description of how benchmark score should or shouldn't be interpreted/used**

- **Explanation:** The benchmark developers provide information about what benchmark users can and cannot take away from the benchmark score.

- **Justification:** Clarifying the interpretation of benchmark scores prevents misuse and misinterpretation, ensuring that users draw accurate conclusions about a model's performance. This guidance helps users apply the scores appropriately within their specific contexts, and understand if the benchmark can be used to assess a model for their desired application context.

- **Points:**
  - 0: The benchmark does not comment on how the benchmark scores should or should not be interpreted.
  - 5: The benchmark acknowledges that the benchmark scores need to be interpreted but gives no guidance on how or how not to do that.

    – 10: The benchmark describes how scores should or should not be interpreted or used, but not both.
    – 15: The benchmark describes how scores should and should not be interpreted or used.

8. **Informed choice of performance metric(s)**

   - **Explanation:** The developers describe how the performance metric for the defined benchmark task should be interpretable, meaningful, and standard for the task thats being evaluated [34]. If a non-standard metric is selected, they describe their rationale for choosing a non-standard metric.

   - **Justification:** The metric should be easily understood by the reader to build their own opinion about the model's capabilities, given the benchmark score. If a non-standard metric is used, an explanation is necessary to clarify its relevance and ensure that users can accurately interpret the results. [34]

   - **Points:**
     – 0: The benchmark does not mention an evaluation metric or does not explain the choice of metric.
     – 5: The benchmark acknowledges the need for an informed metric choice but does not justify their metric choice.
     – 10: The benchmark provides an explanation for the choice of some but not all of their metrics.
     – 15: The benchmark provides an explanation for the choice of all of their metrics.

9. **Includes floors and ceilings for metric**

   - **Explanation:** The benchmark provides clear floors and ceilings for the metric(s) it uses [34].

   - **Justification:** Establishing clear floors and ceilings for metrics ensures that users have a reference point for understanding model performance. It helps users understand if a benchmark is already saturated or if progress can be made on the task [34]. This also allows benchmark developers to decide when a benchmark should be retired.

   - **Points:**
     – 0: The benchmark does not provide any metric floors or ceilings.
     – 5: Floors and ceilings are shown in the results figure but not explicitly mentioned in the text.
     – 10: The benchmark provides floors and ceilings for some but not all evaluation metrics.
     – 15: The benchmark provides floors and ceilings for all evaluation metrics.

10. **Includes human performance level**

    - **Explanation:** The benchmark explicitly states human performance measured on the benchmark task [34]. It also explains how human performance was measured and if this was the performance of an average or expert group of humans. The benchmark notes if measuring human performance is not possible on the benchmark task and why.

    - **Justification:** Similar to the previous criteria, including human performance on a benchmark allows the reader to put the models performance into perspective and allows for a better interpretability of the benchmarking score [34].

    - **Points:**
      – 0: The benchmark does not state human performance and does not explain why this is not applicable here.
      – 5: The benchmark mentions human performance in passing but does not provide a measurement or explanation.
      – 10: The benchmark states human performance but does not explain how it was obtained.

- 15: The benchmark states human performance and explains how it was obtained.
- n/a: The benchmark task cannot be completed by a human, and hence reporting human performance is not possible.

11. **Includes random performance level**

   - **Explanation:** The developers explicitly states the random performance measured on the benchmark [34].
   - **Justification:** By establishing a baseline performance level achieved through random guessing, generation, or selection, benchmark users can better understand the extent to which a model's performance stems from its inherent capabilities, rather than mere chance or the benchmarks design and especially metric choices. This random performance level serves as a reference point, allowing for a clearer assessment of the model's true effectiveness in tackling the specific task at hand.
   - **Points:**
     - 0: The benchmark does not state random performance and does not explain why this is not applicable here.
     - 5: The benchmark mentions random performance but does not provide quantitative random performance on the benchmark task(s).
     - 10: The benchmark states random performance for some but not all tasks.
     - 15: The benchmark states random performance for all tasks.
     - n/a: Measuring random performance on the benchmark task is not possible, and hence reporting random performance is not applicable.

12. **Addresses input sensitivity**

   - **Explanation:** The benchmark contains multiple input variations with the same semantic meaning/intended to elicit the same response or output by the tested model. The developers describe all relevant details such as how many different variations were tested per prompt, and how the variations were designed. For language models, this would mean including a variety of semantically (but not syntactically) equivalent prompts to combat prompt sensitivity [73, 42, 55, 72]. For computer vision models, this could mean inputting a normal, a blurred, and a cropped version of the same image, etc.), while for reinforcement learning, this could mean measuring the sensitivity of learned policies to input features [56].
   - **Justification:** Addressing input sensitivity in a benchmark ensures that the model's performance is consistent across semantically equivalent inputs, thus validating its robustness. Including multiple variations per input and detailing their design allows for inspection and replicable evaluation of the model's capabilities. This serves the goal of approximating intrinsic model capabilities or harms better rather than just measuring "an artifact" [61] of your input.
   - **Points:**
     - 0: The benchmark does not mention or address input sensitivity.
     - 5: The benchmark mentions the issue of input sensitivity but does not describe experiments to test for it.
     - 10: The benchmark includes some input variations with the same semantic meaning but lacks thorough descriptions or details on the number of variations and their design.
     - 15: The benchmark contains multiple input variations with the same semantic meaning, providing detailed descriptions of all relevant details such as the number of variations per prompt and how they were designed.

13. **Validated automatic evaluation available**

   - **Explanation:** Evaluating a model against a benchmark does not require human evaluation in the process and the quality of the automated evaluation is validated (if applicable, e.g., in the case of FM-based evaluations).

- **Justification:** Requiring human feedback to evaluate performance on a benchmark will significantly limit the scalability of the benchmark and potentially introduce biases from the human evaluators themselves. In addition, this may require an IRB for researchers, and will be more costly than an automatic evaluation, creating "major barriers to entry" [34].
- **Points:**
  - 0: The benchmark does not provide any form of automatic evaluation and relies entirely on human evaluation.
  - 5: The benchmark mentions the benefits of automatic evaluation but provides no or limited automatic valuation.
  - 10: The benchmark includes an automatic evaluation method but does not offer any validation.
  - 15: The benchmark includes an automatic evaluation method and describes how it was validated as well as the results of the validation.

14. **Explanation of differences to related benchmarks**

- **Explanation:** The benchmark developers explain how their benchmark fills a gap compared to existing benchmarks or how it expands on existing benchmarks or their tested concepts.
- **Justification:** Benchmark developers demonstrate the added value and relevance of the new benchmark, justifying its necessity by addressing specific gaps in existing benchmarks or by expanding on saturated benchmarks. This allows users to better understand the differences between related benchmarks and determine which one to use for their specific evaluation context.
- **Points:**
  - 0: The benchmarks do not explain any differences or relevance to existing benchmarks.
  - 5: The benchmark briefly mentions existing benchmarks but provides no explanations of differences or added value.
  - 10: The benchmark provides an explanation of how it fills a gap or expands on existing benchmarks for some but not all mentioned related benchmarks.
  - 15: The benchmark provides an explanation of how it fills a gap or expands on existing benchmarks for all mentioned related benchmarks.

### K.2 Benchmark Implementation

1. **Availability of evaluation code**

- **Explanation:** The benchmark developers make the code available for others to evaluate their own models against the benchmark, e.g., as part of a GitHub repository.
- **Justification:**
- **Points:** Without access to the benchmarking procedure itself, the benchmark cannot be scrutinized by external parties to verify its reliability and adequacy, nor can it be utilized for independent evaluations and comparisons by benchmark users. In addition, if benchmark users have to write their evaluation code from scratch, its more likely that seemingly minor implementation details affect the measured performance, hindering a fair comparison [13].
  - 0: The evaluation code is not publicly available.
  - 5: The benchmark mentions the availability of evaluation code but does not provide access to it.
  - 10: The evaluation code is publicly available for some metrics described by the benchmark.
  - 15: The evaluation code is publicly available for all metrics described by the benchmark.

2. **Script to replicate results is explicitly included**

   - **Explanation:** The benchmark developers give access to the input, output, and evaluation code, as well as all other necessary information (e.g., hyperparameters or random seed set) that they used to create the initial benchmarking results presented in the paper.

   - **Justification:** Providing access to the input, output, and code allows for transparency and reproducibility of the reported results, fostering trust into the benchmark, and contributing to overcome the current reproducibility crisis in AI/ML research [35].

   - **Points:**
     - 0: The developers do not provide a script to reproduce the results.
     - 5: The issue of result replicability is mentioned in the benchmark paper but not addressed.
     - 10: A script to reproduce some results in the benchmark paper is available.
     - 15: A script to reproduce all results in the benchmark paper is available.

3. **Accessibility of evaluation data, prompts, or dynamic environment**

   - **Explanation:** The benchmark developers make the evaluation data, prompts, or the data/environment generation mechanism accessible. These do not have to be made public in order to earn full points (if contamination is a concern, for example), but some access to it for evaluation purposes, e.g., by hosting it privately on Hugging Face, needs to be possible.

   - **Justification:** Without any accessibility of the evaluation data, prompts, or environment generation mechanism, a benchmark cannot be used.

   - **Points:**
     - 0: No access to evaluation data, prompts, or data/environment generation mechanism is provided.
     - 5: The existence of evaluation data, prompts, or data/environment generation mechanism is mentioned, but no concrete access is provided.
     - 10: Partial access to evaluation data, prompts, or data/environment generation mechanism is provided, allowing for limited evaluation.
     - 15: Full access to evaluation data, prompts, or data/environment generation mechanism is provided, enabling comprehensive evaluation.

4. **Supports evaluation of models via API calls**

   - **Explanation:** The benchmark developers allow the benchmark evaluation of models via API access, if applicable.

   - **Justification:** This criteria is dependent on the subfield. In NLP, for example, closed-source models such as GPT-4 are oftentimes only accessible via API. Without support for API evaluation, they cannot be evaluated, which is especially problematic if such models are the state-of-the-art models in the field.

   - **Points:**
     - 0: The benchmark does not support evaluation of models via API calls.
     - 5: The benchmark mentions the possibility of API evaluation but does not provide concrete implementation details.
     - 10: The benchmark supports evaluation of models via one API.
     - 15: The benchmark supports evaluation of models via two or more APIs to different models.

5. **Supports evaluation of local models**

   - **Explanation:** The benchmark developers implement code to support the evaluation of local models without API access.

   - **Justification:** Some model developers only host their models locally. A benchmark should support the evaluation of those to allow for a wide variety of models to be evaluated against the benchmark.

- **Points:**
  - 0: The benchmark requires users to write their own code to evaluate a local model.
  - 5: The benchmark mentions that local evaluation should be possible but doesn't provide corresponding code.
  - 10: The benchmark provides minimal support for local model evaluation, requiring significant user effort.
  - 15: The benchmark provides full support for local model evaluation with user-friendly code.

6. **Inclusion of a globally unique identifier or encryption of evaluation instances**
   - **Explanation:** Benchmark developers include a globally unique identifier (GUID) or canary string in the main public evaluation code and all public evaluation prompt or data files. Alternatively, they encrypt the test data files and make the key public.
   - **Justification:** Including a GUID in relevant (sub-)repositories, public code and data repositories can support the identification of data contamination in models [74], either by allowing model developers to filter out the evaluation data out of large amounts of web-scraped data or by allowing benchmark developers to identify which model developers trained on their data and hence have created models that potentially perform better than they would otherwise on the benchmark. Encrypted test data files prevent non-adversarial crawling of such data; however, [36] advise against "using standard obfuscation or compression methods that are not key-protected, since some crawling systems include pipelines of automatic decompression or deobfuscation."
   - **Points:**
     - 0: The benchmark does not include a GUID or encryption of evaluation instances.
     - 5: The benchmark acknowledges the risk of contamination but does not address it.
     - 10: The benchmark partially implements a GUID or encryption, but not consistently across all relevant files.
     - 15: The benchmark consistently includes a GUID or encryption across all relevant files and repositories.

7. **Inclusion of 'training_on_test_set' task**
   - **Explanation:** The benchmark includes a task to identify if the model was trained on the benchmark data.
   - **Justification:** Public benchmarks face the challenges that their evaluation data may be web-scraped and used to train a model. A 'training_on_test_set' task can serve as a "post-hoc diagnosis of whether [... benchmark] data was used in model training." [74]
   - **Points:**
     - 0: The benchmark does not include a 'training_on_test_set' task.
     - 5: The benchmark mentions the possibility that models were trained on its data but does not provide a way to check it.
     - 10: The benchmark includes a partial or limited implementation of a 'training_on_test_set' task that only tests for part of the data used.
     - 15: The benchmark includes a comprehensive 'training_on_test_set' task.

8. **Assess need for warnings for sensitive/harmful content**
   - **Explanation:** Benchmark developers explicitly mention in the paper if the evaluation tasks or the expected output may contain sensitive or harmful content. If they do not anticipate sensitive/harmful content in either case, they should explicitly state that.
   - **Justification:** By explicitly stating the presence of sensitive or harmful content and issuing appropriate warnings, developers help users make informed decisions and take necessary precautions. Even if developers do not expect sensitive or harmful content, if they state that, they showcase to the benchmark users that they actually thought about the possibility. Otherwise, users couldn't be sure if the input or output doesn't contain problematic content or if the developers just forgot to include a warning.

- **Points:**
  - 0: The benchmark does not mention that they checked for the presence or absence of sensitive/harmful content in the evaluation tasks or expected output.
  - 5: The benchmark mentions the general possibility of sensitive/harmful content but does not provide clear statements or warnings.
  - 10: The benchmark explicitly states the presence or absence of sensitive/harmful content for either the evaluation tasks or the expected output.
  - 15: The benchmark explicitly states the presence or absence of sensitive/harmful content for both the evaluation tasks and the expected output.

9. **Release requirements specified**

   - **Explanation:** Benchmark developers specify rules for benchmark users to "ensure the integrity of test results" [82]. While not all benchmark developers will be able to enforce the release requirements, they should at least specify them. One example is: "1. Publishers do not train directly on or against the benchmark dataset and retract any reported results if and when benchmark data is found to have been in training data. 2. Techniques that are likely to increase the test performance without a commensurate increase in safety factor are discouraged and may result in benchmark exclusion. [...]" [82]

   - **Justification:** Written terms of use can help to set expectations and have a foundation to address subsequent contamination or intentional gamification attempts of the benchmark. Potential options they could mention in case of release requirement breaches are, e.g., "publishing public statements correcting the public record" or "resulting in the [model] being permanently banned from the benchmark" [82]; however, we will not assess the enforcement ability or potential listed sanctions as part of this criteria, just the statement of release requirements.

   - **Points:**
     - 0: The benchmark does not specify any release requirements for benchmark users.
     - 5: The benchmark briefly mentions the issue of potential gameability or misuse by benchmark users but does not provide specific details.
     - 10: The benchmark states dos and donts how to use the benchmark but does not specify these as requirements for use.
     - 15: The benchmark provides a set of release requirements for benchmark users.

10. **Includes *Build Status* or equivalent**

    - **Explanation:** A build status is a feature, typically implemented as a GitHub Action, that indicates whether the most recent build of the benchmark was successful [28]. It should be implemented for the benchmark's evaluation code. It verifies that the code is running correctly after the latest commit.

    - **Justification:** A passing build status signifies that the main evaluation code was usable at the latest commit [28]. Including a build status or equivalent can help to ensure the reliability and usability of the evaluation code. It allows benchmark users to quickly determine if the code is functioning as intended, saving time and effort in identifying potential issues.

    - **Points:**
      - 0: The benchmark neither references nor implements any form of build status or equivalent.
      - 5: The benchmark mentions the need for working evaluation code but does not implement it in any meaningful way.
      - 10: The benchmark partially implements a build status or equivalent by providing the information in a less accessible manner.
      - 15: The benchmark fully implements a build status or equivalent, clearly displaying the status of the most recent build and providing easy access to the information.

### K.3 Benchmark Documentation

1. **Requirements file available**

    - **Explanation:** A requirements or environment file, or equivalent is available.
    - **Justification:** Ease of use is a key criteria for benchmark adoption. Providing a requirements file allows for the quick installation of relevant packages at the correct versions, e.g., within a virtual environment, to use the evaluation code.
    - **Points:**
        - 0: No requirements file or equivalent is provided.
        - 5: A requirements file is mentioned but not provided.
        - 10: A requirements file is provided but may be missing some dependencies or versions.
        - 15: A complete and accurate requirements file specifying all necessary dependencies and versions is provided.

2. **Quick-start guide or demo code available**

    - **Explanation:** The benchmark developers make a quick start guide or demo available that walks step-by-step through how the benchmark can be used.
    - **Justification:** Similar to the criteria above, ease of use is a key criteria for benchmark adoption. Providing a quick-start guide takes away any guesswork on the user side and allows them to directly set up and use the benchmark without spending extra time on setup issues.
    - **Points:**
        - 0: No quick-start guide or demo code is provided.
        - 5: A quick-start guide or demo code is mentioned but not provided.
        - 10: A quick-start guide or demo code is provided but may be missing some steps or details.
        - 15: A comprehensive, step-by-step quick-start guide or demo code is provided.

3. **Includes informative In-line code comments**

    - **Explanation:** In-line code comments state the purpose, inputs, outputs, and functionality of each code segment in all files relevant for the benchmark evaluation.
    - **Justification:** In-line documentation of code enhances clarity, understanding, and reproducibility. It facilitates collaboration, maintainability, and makes debugging easier for benchmark developers and users, should that be necessary.
    - **Points:**
        - 0: No in-line code comments are provided.
        - 5: In-line code comments are sparse and do not adequately explain the purpose, inputs, outputs, or functionality of the code.
        - 10: Informative in-line code comments are present for most of the code but may be lacking in detail or clarity for some code segments.
        - 15: Comprehensive and informative in-line code comments are provided for all relevant code segments, clearly explaining their purpose, inputs, outputs, and functionality.

4. **Code documentation available**

    - **Explanation:** A full documentation of the repository and code it entails is publicly available. This includes, for example, an overview of the folder structure, the files in the repo, an explanation of functions in the repo.
    - **Justification:** Detailed documentation of code enhances clarity, understanding, and reproducibility. It facilitates collaboration, maintainability, and makes debugging easier for benchmark developers and users, should that be necessary.
    - **Points:**

- 0: No code documentation is provided.
- 5: Code documentation is mentioned but not provided.
- 10: Code documentation is minimal or incomplete, lacking important details about the repository structure and functions.
- 15: Comprehensive code documentation is provided, including a clear overview of the folder structure, files in the repo, and detailed explanations of all relevant functions.

5. **Documentation of test task categories & rationale**

- **Explanation:** The benchmark developers define the tasks or task categories a model is tested on and describe the rationale for choosing the tasks or task categories. The rationale should explain how these tasks are relevant to the benchmark's objectives, what they aim to measure, and why they are important for evaluating the concept or capability to be tested.

- **Justification:** Documenting test tasks is essential for transparency and for allowing public scrutiny of the benchmark. The rationale provides insight into the selection process, demonstrating that the tasks are not arbitrary but are carefully chosen to reflect real-world applications and user needs. Both help users decide if the benchmark is adequate for their evaluation contexts.

- **Points:**
  - 0: No documentation of test task categories or rationale is provided.
  - 5: Test task categories are mentioned but they are neither defined in detail and a rationale for their selection is missing or inadequate.
  - 10: Test task categories are defined, but the rationale for their selection is not provided.
  - 15: Test task categories are clearly defined, and a comprehensive rationale is provided, explaining their relevance to the benchmark's objectives, what they measure, and their importance for evaluating the targeted concept or capability.

6. **Documentation of assumptions about normative properties**

- **Explanation:** If the benchmark measures properties that vary across cultural contexts (e.g., politeness), then normative assumptions are explicitly stated. The benchmark developers clearly define the cultural context and values that the benchmark adheres to, explaining how the measured properties are conceptualized and operationalized within the benchmark.

- **Justification:** By explicitly stating normative assumptions, the authors provide transparency about the cultural framework and values that guide the benchmark's design and evaluation criteria, which can subsequently ensure cultural sensitivity and mitigate potential biases. It also facilitates informed decision-making for users of benchmarks, specifically for culture-dependent use cases they're interested in, such as measuring toxicity or bias, for example.

- **Points:**
  - 0: No documentation of normative assumptions is provided, even though the benchmark measures culturally-dependent properties.
  - 5: The potential influence and importance of cultural context on the benchmark is acknowledged but normative assumptions aren't stated.
  - 10: Normative assumptions are stated, but the explanation of how they are conceptualized and operationalized within the benchmark is incomplete or lacks clarity.
  - 15: Normative assumptions are explicitly and clearly stated, defining the cultural context and values that the benchmark adheres to, and explaining how the measured properties are conceptualized and operationalized within the benchmark.

7. **Documentation of limitations**

- **Explanation:** Benchmark developers outline the limitations of the benchmark, including but not limited to the tasks, contexts, and scenarios that are not covered by the evaluation are acknowledged. It's stated which use cases are out-of-scope.
- **Justification:** Documenting a benchmark's limitations is necessary for users to assess its suitability for their specific evaluation needs. By understanding what the benchmark does not cover, users can make informed decisions about whether the benchmark aligns with their goals and whether additional evaluations (either in the form of other benchmarks or private evaluations) may be required to complement the benchmark's results.
- **Points:**
  - 0: No documentation of the benchmark's limitations is provided.
  - 5: Limitations of AI evaluations more broadly are briefly mentioned but without any detail and not applied to the specific benchmark.
  - 10: Either limitations regarding the applicability and use of the benchmark or limitations of the benchmark design are discussed, but not both.
  - 15: Both limitations regarding the applicability and use of the benchmark and limitations of the benchmark design are comprehensively discussed.

8. **Documentation of benchmark construction process**

- **Explanation:** Benchmark developers give a detailed account of the design process, including the specific decisions made at each lifecycle stage, the rationale behind them, and any trade-offs or compromises (e.g., balancing complexity vs. practicality) considered.
- **Justification:** Documenting the benchmark design process is essential for transparency, as it allows users to understand how the benchmark was created and what factors influenced its development. It allows users to assess the thoroughness and rigor of the benchmark's construction. This information further enables users to critically evaluate whether the benchmark is suitable for their specific use case.
- **Points:**
  - 0: No documentation of the benchmark construction process is provided.
  - 5: The benchmark construction process is briefly mentioned but lacks sufficient detail about the decisions made, rationale, and trade-offs considered.
  - 10: The benchmark construction process is documented, including some decisions made and their rationale, but the description lacks depth or fails to address important aspects such as trade-offs or compromises.
  - 15: The benchmark construction process is comprehensively documented, providing a detailed account of the specific decisions made at each stage, the rationale behind them, and any trade-offs or compromises considered.

9. **Provision of a globally unique, persistent identifier for a dataset and its metadata**

- **Explanation:** The benchmark dataset and its associated metadata are assigned a globally unique and persistent identifier, such as a Digital Object Identifier (DOI), to ensure long-term accessibility and citability of the resource (FAIR Principles, 2024).
- **Justification:** A persistent identifier supports the findability and accessibility of the benchmark and its dataset. It allows for unambiguous referencing of the data, facilitates proper attribution, and ensures that the dataset can be located and accessed over time, even if its physical location changes. This practice aligns with the FAIR (Findable, Accessible, Interoperable, Reusable) principles, enhancing the benchmark's scientific value and reusability.
- **Points:**
  - 0: The benchmark paper, dataset, and metadata are not assigned any persistent identifier.

– 5: The benchmark assigns persistent identifiers to the paper, the dataset, or the metadata.
– 10: The benchmark assigns a persistent identifier to two out of three (paper, dataset, metadata).
– 15: The benchmark assigns a globally unique, persistent identifier to the dataset, its metadata, and the paper.

10. **Inclusion of standardized metadata (e.g., following the Croissant standard)**

   • **Explanation:** The benchmark includes comprehensive, standardized metadata that describes the dataset, its structure, and relevant information about its creation and usage. This metadata adheres to established standards such as the Croissant standard, which is designed specifically for machine learning datasets.

   • **Justification:** Standardized metadata is crucial for ensuring interoperability and reusability of the benchmark dataset. It provides consistent and machine-readable information about the dataset's contents, structure, and provenance. This standardization facilitates easier discovery, understanding, and integration of the dataset into various research workflows. By following established standards like Croissant, the benchmark enhances its utility across different platforms and tools in the machine learning ecosystem.

   • **Points:**
     – 0: The benchmark does not include any structured metadata.
     – 5: The benchmark includes some basic metadata, but it is not standardized or comprehensive.
     – 10: The benchmark includes comprehensive metadata that covers most aspects of the dataset, but it does not fully adhere to a recognized standard like Croissant.
     – 15: The benchmark includes complete, standardized metadata (e.g., following the Croissant standard) that thoroughly describes all aspects of the dataset, ensuring maximum interoperability and reusability.

11. **Documentation of data sources and how the data was collected (if applicable)**

   • **Explanation:** The benchmark provides comprehensive documentation detailing the origins of the data, the methods used for data collection, and, where applicable, discusses issues of data provenance and informed consent. They also list the license types for all data used and how they ensured compliance with that license.

   • **Justification:** Thorough documentation of data sources and collection methods is necessary for ensuring transparency, reproducibility, and ethical design of the benchmark. It allows users to understand the context and limitations of the data, assess its appropriateness for their specific use cases, and make informed decisions about its application. Furthermore, discussing data provenance and informed consent addresses ethical considerations, particularly when dealing with sensitive or personal data, and helps ensure compliance with data protection regulations.

   • **Points:**
     – 0: The benchmark provides no information about data sources or collection methods.
     – 5: The benchmark mentions data sources but provides minimal details about collection methods or ethical considerations.
     – 10: The benchmark includes a detailed description of data sources and collection methods, but lacks a discussion of data provenance, compliance with licensing, or informed consent, where applicable.
     – 15: The benchmark provides extensive documentation of data sources, collection methods, and a thorough discussion of data provenance, compliance with licensing, and informed consent, addressing relevant ethical and legal considerations.

12. **Documentation of the data preprocessing steps taken**

- **Explanation:** The benchmark provides a detailed account of all preprocessing steps applied to the raw data before its inclusion in the final dataset. This documentation includes information on data cleaning, normalization, feature engineering, handling of missing values, and any other transformations or manipulations performed on the original data. If no data preprocessing was done, the authors state this explicitly.
- **Justification:** Thorough documentation of preprocessing steps is necessary for ensuring reproducibility and transparency of the benchmark. It allows users to understand exactly how the final dataset was created, which is key for interpreting results, replicating experiments, and assessing the benchmark's applicability to different use cases. Additionally, this information helps identify potential biases or artifacts introduced during preprocessing that could affect model performance or generalization.
- **Points:**
  - 0: The benchmark provides no information about data preprocessing steps.
  - 5: The benchmark mentions that preprocessing was done but offers minimal details about the specific steps taken.
  - 10: The benchmark includes a general description of preprocessing steps, but lacks comprehensive details or fails to cover all aspects of the data preparation process.
  - 15: The benchmark provides an exhaustive, step-by-step documentation of all preprocessing procedures, including rationales for choices made and potential impacts on the data.

13. **Documentation of the data annotation process (if applicable)**

- **Explanation:** The benchmark provides documentation of the data annotation process, including the annotation guidelines, the qualifications and training of annotators, the annotation tools used, quality control measures, and inter-annotator agreement metrics. This documentation covers the entire workflow from raw data to the final annotated dataset.
- **Justification:** Comprehensive documentation of the annotation process is necessary for understanding the quality, reliability, and potential biases in the labeled data. It allows users to assess the suitability of the dataset for their specific tasks and to interpret results accurately. Transparent annotation documentation also enables reproducibility of the labeling process, facilitates improvements in future iterations of the benchmark, and helps in identifying and mitigating potential sources of bias or error in the annotations.
- **Points:**
  - 0: The benchmark provides no information about the data annotation process.
  - 5: The benchmark mentions that data was annotated but offers minimal details about the process or guidelines used.
  - 10: The benchmark includes a general description of the annotation process, including guidelines and tools used, but lacks comprehensive details on quality control measures or inter-annotator agreement.
  - 15: The benchmark provides exhaustive documentation of the entire annotation process, including detailed guidelines, annotator information, quality control measures, inter-annotator agreement metrics, and discussions of potential biases or limitations in the annotation approach.

14. **Documentation of the representativeness of the data (if applicable)**

- **Explanation:** The benchmark provides analysis and documentation of how representative the dataset or environment is of the target population or domain. This includes an explanation of the sampling procedure used, any potential biases in the data collection process, and how well the dataset captures the diversity and distribution of the intended population or phenomenon being studied.
- **Justification:** Understanding the representativeness of the data is necessary for assessing the generalizability and validity of any conclusions drawn from models trained

or evaluated on the benchmark. It helps users identify potential limitations or biases in the dataset that could affect model performance in real-world applications. Proper documentation of representativeness also aids in interpreting benchmark results within the context of the population it represents and highlights areas where the dataset may need expansion or improvement to better cover underrepresented groups or scenarios.

- **Points:**
  - 0: The benchmark provides no information about the representativeness of the data or the sampling procedure used.
  - 5: The benchmark mentions the importance of data representativeness but offers minimal analysis or explanation of how representative the dataset actually is.
  - 10: The benchmark includes a general discussion of data representativeness and the sampling procedure, but lacks comprehensive analysis or fails to address potential biases or limitations in representativeness.
  - 15: The benchmark provides an in-depth analysis of data representativeness, including detailed explanation of the sampling procedure, quantitative measures of population coverage, discussion of potential biases, and acknowledgment of any limitations in representativeness.

15. **Standardized documentation**

- **Explanation:** The benchmark utilizes a standardized documentation format, such as data cards, to present the information about the dataset that is underlying to the benchmark. This standardized approach ensures that all key aspects of the dataset are systematically covered, including its composition, collection methodology, intended uses, ethical considerations, and potential biases.

- **Justification:** Adopting a standardized documentation scheme like data cards enhances the usability and transparency of the benchmark. It provides a consistent, structured format that makes it easier for users to quickly understand the dataset's characteristics, limitations, and appropriate use cases. Standardized documentation facilitates easier comparison between datasets and benchmarks, promotes best practices in data reporting, and helps identify potential issues or gaps in the dataset's coverage.

- **Points:**
  - 0: The benchmark does not use any standardized documentation scheme.
  - 5: The benchmark includes some elements of standardized documentation, but does not fully adhere to an established scheme like data cards.
  - 10: The benchmark uses a standardized documentation scheme, but some sections are incomplete or lack detail.
  - 15: The benchmark fully implements a comprehensive standardized documentation scheme (e.g., data cards), providing thorough and structured information on all relevant aspects of the dataset.

16. **Documentation of evaluation metric(s)**

- **Explanation:** The evaluation metrics used are clearly specified and defined, both for standard and custom metrics tailored to the specific task or domain. The exact formulas or processes used to calculate these metrics, along with any parameters or thresholds employed, are made transparent.

- **Justification:** Documenting the evaluation metrics and scoring process is essential for enabling users to understand how the benchmark quantifies model performance and determines rankings or comparisons. By providing clear and detailed information about the metrics and scoring methods, users can assess whether the chosen metrics are appropriate for the task at hand, align with their own evaluation criteria, and provide a fair and meaningful basis for comparing different models or approaches.

- **Points:**
  - 0: No documentation of the evaluation metrics is provided.

- 5: The evaluation metrics are mentioned but not clearly defined, and the exact formulas or processes used to calculate them are not provided.
- 10: The evaluation metrics are defined, but the documentation lacks some important details, such as any parameters or thresholds employed.
- 15: The evaluation metrics are clearly specified. The exact formulas or processes used to calculate these metrics, along with any parameters or thresholds employed, are comprehensively documented.

17. **Report statistical significance of benchmark results for at least one model**

- **Explanation:** Benchmark developers run statistical significance tests on the benchmark results. They report results for, e.g., more than one random seed, and provide variance bounds. In cases where the benchmark is perfectly deterministic, this should be explicitly stated.

- **Justification:** Not doing statistical significance testing can significantly reduce the validity, utility and confidence in results [13]. Especially for benchmarks, we want to understand how much of the results are due to noise and how much is caused by true differences between the models tested.

- **Points:**
  - 0: No statistical significance testing or variance reporting is provided for the benchmark results.
  - 5: The need for valid benchmarks and/or statistical significance or uncertainty estimation is mentioned but not not addressed.
  - 10: Benchmark developers if "bound the expected variation across model training runs" [40], [13]
  - 15: Benchmark developers run statistical significance tests on the benchmark results for at least one model and provide variance bounds or other uncertainty estimations. In cases where the benchmark is perfectly deterministic, this is explicitly stated.

18. **Accepted at peer-reviewed venue**

- **Explanation:** The benchmark/its associated paper was accepted to a peer-reviewed journal, conference, or similar venue.

- **Justification:** Acceptance at a peer-reviewed venue signifies that the benchmark has undergone an evaluation by an external party, ensuring its validity, reliability, and scientific merit [5]. This peer review process contributes to the credibility and assurance to users that the benchmark meets established standards of quality and relevance [5].

- **Points:**
  - 0: The benchmark/its associated paper has not been accepted at a peer-reviewed venue.
  - 5: The benchmark/its associated paper has been submitted to a peer-reviewed venue but is still under review or awaiting acceptance.
  - 10: The benchmark/its associated paper has been accepted at a peer-reviewed workshop or symposium.
  - 15: The benchmark/its associated paper has been accepted at a peer-reviewed journal, conference, or similar high-profile venue.

19. **Specifies applicable license**

- **Explanation:** The benchmark developers clearly specify the applicable license for the benchmark in the code repository or paper. This includes providing information about the conditions under which the benchmark can be used, modified, and distributed.

- **Justification:** Specifying the applicable license ensures legal clarity and compliance for benchmark users and enables wider adoption, as commercial users might not be able to use the benchmark if no license is specified.

- **Points:**

- 0: No license is specified for the benchmark.
- 5: A license is mentioned but not clearly specified or linked to in the code repository or paper.
- 10: A license is specified but lacks some important details about the conditions under which the benchmark can be used, modified, or distributed.
- 15: The applicable license for the benchmark is clearly specified in the code repository or paper, providing comprehensive information about the conditions under which the benchmark can be used, modified, and distributed.

### K.4 Benchmark Maintenance

1. **Code usability checked within the last year**

   - **Explanation:** The main files of the public code were updated within the last year[8], or the developers checked that the benchmark code is still usable and explicitly state this check in the README file, including the date of the check.
   - **Justification:** Over time, packages that the benchmark depends on may be updated and become incompatible with the original evaluation/benchmark code. To ensure ongoing usability, benchmark developers must check if their code can still be used at least once a year[9]. This practice ensures that users can use the benchmark without encountering and having to fix issues due to outdated dependencies.
   - **Points:**
     - 0: No updates to the main files of the public code within the last year, and no explicit statement of a usability check in the README file.
     - 5: Updates to minor files in the repo were made (e.g., README file) but an explicit statement of a usability check in the README file is not reported.
     - 10: Updates to the main files of the public code were made within the last year, but the build status check failed and wasn't fixed.
     - 15: Updates to the main files of the public code within the last year, accompanied by a successful build status check, or an explicit statement of a usability check in the README file, including the date of the check was provided.

2. **Maintained feedback channel for users**

   - **Explanation:** GitHub issues are acknowledged or addressed within three months. If there are no open issues, benchmark developers would get full points.
   - **Justification:** Over time, users may find issues with the benchmark tasks or implementation. To ensure continued usability, benchmark developers should address these concerns in a reasonable amount of time. Promptly responding to user feedback helps maintain the reliability and relevance of the benchmark.
   - **Points:**
     - 0: No acknowledgment or response to GitHub issues that are older than three months[10].
     - 5: GitHub issues are mentioned as a way to provide feedback but there are GitHub issues that were not responded to and that are older than three months.
     - 10: All GitHub issues are acknowledged within three months, but not all are addressed or resolved or were closed because the issue/feature request won't be attended to.

---

[8]We recognize that this criterion is just a proxy for checking code usability, but we assume that if the main code was edited and a build status [28] passed, that the usability was sufficiently checked.

[9]The one-year threshold is somewhat arbitrary but out of experience of the authors, there is some transition period until which old versions can still be reliably used and are maintained, which can vary from a few months to a few years.

[10]This is an arbitrary cut-off time but it seemed reasonable to give developers extended time to respond to open issues.

– 15: All GitHub issues are acknowledged and addressed within three months, or it is clearly stated if an issue cannot be fixed or if a feature request won't be fulfilled. Alternatively, there are no open issues[11].

3. **Provide contact details of person responsible for benchmark**

   - **Explanation:** The benchmark should include contact details of the person responsible, such as a corresponding author in the associated paper, a contact person listed on GitHub or the website, or an available online feedback form.
   - **Justification:** Providing contact details ensures that users have a communication channel for inquiries, feedback, or reporting issues related to the benchmark. This transparency supports effective collaboration and resolution of problems, enhancing the benchmark's usability.
   - **Points:**
     - 0: It is not disclosed who developed the benchmark.
     - 5: The benchmark developers are disclosed but no explicit contact details are provided.
     - 10: Contact details are provided but are incomplete or difficult to find, e.g., only as part of terms of service on a website.
     - 15: Contact details of the person responsible for the benchmark are easily accessible, such as a corresponding author in the associated paper, a contact person listed on GitHub or the website, or an available online feedback form.

---

[11]This is an imperfect proxy for a maintained feedback channel. It may be that the benchmark is working well or it may be that the benchmark is not used enough for issues to occur. However, maintenance is a critical part of benchmarks, and we hence decided to include an imperfect proxy rather than not including this criterion at all.

