# OpenReview forum: "BetterBench: Assessing AI Benchmarks, Uncovering Issues, and Establishing Best Practices"
_NeurIPS.cc/2024/Datasets_and_Benchmarks_Track — NeurIPS 2024 Track Datasets and Benchmarks Spotlight_

### Official Review · Reviewer_2Wd9 · 2024-07-21
**A comprehensive framework for evaluating benchmarks**

**Rating:** 7
**Confidence:** 5
**Correctness:** Yes.
**Clarity:** Yes.

**Review:**

With so many benchmarks available, this paper provides a significant contributions to build and assess usable benchmarks.  The quality of the paper is good and the work seems comprehensive.

**Strengths:**

-A set of 40 criteria based on expert interviews and literature survey
-Evaluation of existing benchmarks, including both foundation and non-foundation models
-A playbook for developers to improve benchmark quality and a repository for continuous updates

**Additional Feedback:**

None.

**Documentation:**

Yes.

**Ethics:**

no.

**Opportunities For Improvement:**

The authors could present or call for tools to implement the ideal benchmark and its assessment.

**Relation To Prior Work:**

Yes.

**Summary And Contributions:**

The paper introduces an assessment framework to evaluate the quality of AI benchmarks called BetterBench.  The contributions include a comprehensive approach to evaluation benchmarks across AI life cycle. The paper also includes a checklist for  quality metrics.

---

> ### Author Rebuttal · Authors · 2024-08-16
>
> Dear reviewer 2Wd9,
>
> Thank you for your positive feedback! You mention as opportunities for improvement that “The authors could present or call for tools to implement the ideal benchmark and its assessment.”. We are happy to include a call to implement our checklist as standard for benchmarks developers to ensure a minimum quality standard of published benchmarks.
>
> Given the mix of quantitative and qualitative criteria in our assessment framework, there is a lot of nuance in benchmark development that cannot be automated entirely in a toolkit yet unfortunately. One idea would be to create a standardized, structured metadata format for benchmarks that can at least be automatically parsed by benchmark users and consumers for subsequent analysis. We’re happy to add this as a potential direction for future work to our paper. We will also put this on our to-do list, given that we plan to maintain and add to our database in the future, and make this effort as useful as possible to the community to contribute to raising AI benchmark standards. Would this address your improvement suggestions? Do you see any other improvement suggestions for our work?
>
> Thank you so much for your time and feedback!

---

> > ### Author Response · Authors · 2024-08-27
> >
> > Dear reviewer 2Wd9, given that we’re more than halfway through the author-reviewer discussion period, we wanted to send a friendly reminder that we’d love to hear your thoughts on our rebuttal! Please let us know if there are any remaining questions or concerns. If we addressed your previous concerns sufficiently, we’d be delighted if you’d consider raising your final score. Thank you so much again for your time and feedback!

---

### Official Review · Reviewer_fLHh · 2024-07-25
**Useful benchmark evaluation rubric, but curious lack of data curation/documentation considerations**

**Rating:** 7
**Confidence:** 3
**Correctness:** Yes, to my knowledge, the claims made…
**Clarity:** Yes, the paper is well written with m…

**Review:**

This paper presents high-quality work in assessing AI/ML benchmark practices, and the rubric and checklist proposed by the authors have the potential to be useful tools for the community.

The submission is written and organized well and the methodology is, for the most part, clearly described. However, a major drawback is the omission of a discussion of dataset documentation practices and data curation considerations, both of which are likely to be foundational aspects of a benchmark’s usability. This is discussed in more detail later in the review.

**Strengths:**

- The authors’ proposed assessment rubric has the potential to be a very useful tool for the ML community, guiding new benchmarks toward best practices.
- The authors did not only consult with AI/ML researchers during their development process; they also consulted with policymakers and model users (presumably not AI/ML experts) to get a diversity of perspectives on the considerations of downstream users.
- The authors clearly explain their methodology and rubric through excellent use of figures.

**Additional Feedback:**

The authors should be commended for this work. However, I have scored it below the acceptance threshold for now as I cannot get past the omission of data curation considerations and documentation frameworks from the assessment framework and the references. To update the score to recommend acceptance, I would require revisions to the paper or further justification from the authors on this point.

**Documentation:**

The dataset is not currently available for reviewer access, but the authors have pledged to host and maintain the data via a GitHub repository and have licensed it with a CC BY 4.0. However, the benchmark lacks dataset documentation via a framework (e.g., a datasheet, data card, data statement, or nutrition label) and lacks a Croissant metadata record; both of which are recommended in the NeurIPS D&B call for papers.

**Ethics:**

No, I do not think the paper warrants further ethical discussion or review.

**Limitations:**

Yes, the authors have adequately addressed the limitations and impacts of their work in Sections 8 and 9.

**Opportunities For Improvement:**

- Section 3, Step 1: It is commendable that the authors garnered feedback from stakeholders across these various groups to inform their benchmark lifecycle model. However, it would be nice to have more details on this process and what the feedback looked like. For example, were these representatives asked to map the benchmark life cycle? Or to propose requirements desired of benchmarks? Were they shown the proposed life cycle model by the authors? Did all representatives agree with the final life cycle model? It is mentioned in Step 3 that an initial assessment was shown to the stakeholders: did all stakeholders’ feedback agree? How were disagreements resolved?

- Section 4.2: Assuming that the step in which a benchmark is made publicly available is part of the implementation stage, shouldn’t this stage contain data curation criteria? E.g., the FAIR principles, which lay out recommendations for best practices for sharing datasets. For a recent overview of data curation principles translated for the ML community, see https://doi.org/10.1145/3630106.3658955. I would imagine that these criteria in this section ought to be expanded to include some of these important ideas as well, such as providing a persistent identifier for a dataset and its metadata (e.g., via a DOI). Perhaps some benchmarks are compiling existing datasets, but many of the benchmarks submitted to this track, for example, are introducing new datasets and should have these considerations in mind.

- Section 4.3: Criteria #10 appears to be the only criterion that addresses data documentation, when this is a large area of discussion in ML. The paper I referenced in the last point constructs a rubric for assessing a dataset’s documentation along several data curation axes. It’s quite curious that dataset documentation is not a larger part of this assessment framework, despite the importance of datasets to the definition of a benchmark provided by the authors in lines 23-26.

- Figure 6: Following up on the previous point, I wonder if the lack of thorough scoring on dataset documentation also contributes to benchmarks’ high performance in the Documentation stage. Also, minorly, this figure does not appear to be referenced in the inline text.

**Relation To Prior Work:**

Related work is discussed in Section 2 and scattered throughout the paper. To my knowledge, this discussion is largely satisfactory and differentiates this contribution from previous work. However, as discussed above, I think the relationship between benchmark quality and the body of literature on data documentation in ML should be discussed in this paper (e.g., datasheets, or any of the recommended documentation frameworks cited in the NeurIPS D&B Call for Papers).

**Summary And Contributions:**

This paper develops a benchmark assessment rubric for evaluating how well a benchmark adheres to best practices throughout the benchmark lifecycle. The authors apply the framework to 40 benchmarks and demonstrate that several popular benchmarks suffer from various issues.

---

> ### Author Rebuttal · Authors · 2024-08-16
>
> **Overall:** Dear reviewer fLHh, we genuinely appreciate your detailed, constructive review. We will address your feedback below, but the bottom line is that we agree with you on all points made and have proposed improvement suggestions to our paper to address these concerns. Your feedback already made our paper significantly better and we appreciate you for that!
>
> **Clarification of Methodology:** We'll add more details to our methodology, including a flow diagram (see attached PDF) similar to Fig. 1 in the paper you referenced. Our structured input aimed to understand what information benchmark users/consumers would need to find benchmarks informative and usable. Stakeholders did not disagree with presented requirements; they mainly suggested additions and nuances (e.g., positionality statement) or different ways to operationalize the proposed requirements. We incorporated newly proposed requirements and moved less definitive criteria to "Other Design Considerations", if stakeholders flagged that a previously proposed requirement or design consideration was context dependent. The resulting lifecycle model and requirements list were iteratively defined based on these inputs, though not fully revalidated by all original stakeholders. We'll clarify these points and provide additional details in the final write-up.
>
> **More Data Curation & Documentation Criteria:** You are right, we should have dedicated more specific criteria to data curation and documentation. We discussed this point during our research but initially decided to not include them for two reasons: 1) In our conversations with stakeholders, we heard the recurring theme of “there are reasonably well-established standards and guidelines for developing datasets but not for benchmarks specifically”, and we wanted to fill that specific gap with our work, and 2) we did not want to create another set of guidelines for datasets when there is already great work on the topic out there, to avoid confusion, and expecting that benchmark developers would just use existing dataset curation & documentation standards. However, you are right that we a) did not make this expectation transparent and b) that it is better to have all guidelines for benchmarks in one place, and build on existing work. What we should have done is just incorporate existing dataset frameworks into our assessment and point our readers to the great work that has been previously done on the topic. We agree with you that, if we hope for this to become a minimum standard for benchmarks, this has to involve more detailed data curation and documentation requirements.
>
> Fortunately, there has already been some overlap between our framework and the paper you highlighted (e.g., documentation of scope, real-world usage, involvement of domain experts, …). To the extent that we missed more detailed data curation and data documentation best practices (mainly based the paper you suggested, the FAIR principles, data cards, and Neurips guideline suggestions), we will add the following criteria and re-score benchmarks based on the new list prior to the publication of our paper:
>
> - Provision of a globally unique, persistent identifier for a dataset and its metadata (e.g., via a DOI) (as suggested by you)
> - Inclusion of standardized metadata (e.g., following the Croissant standard)
> - Documentation of data sources and how the data was collected (if applicable) (Discussion of data provenance/informed consent)
> - Documentation of the data preprocessing steps taken
> - Documentation of the data annotation process
> - Documentation of the representativeness of the data (if applicable) (Explanation if “sampling procedure is representative of the population”)
> - Standardized documentation (Dataset documentation follows a standard scheme such as data cards)
> - Harms and benefits analysis (Documentation of discussion on harms and benefits of the benchmark creation)
>
> We hope this addresses your concern; if not, please let us know if there are other criteria we are still missing and we’d be happy to incorporate them!
> We would further kindly ask for your support during the author/reviewer discussion period. Reviewer X3U4 expressed concerns that we are setting "unrealistically high standards," and we expect that the inclusion of additional criteria will increase this concern on their end. We believe in encouraging benchmark developers to strive for better practices rather than lowering standards, and hence would appreciate it if you could weigh in here. Thank you so much!
>
> **Metadata & Prior Work:**
>
> This is another good point (thank you again for such a constructive review!). We'll add a paragraph to the related work section clarifying our work's relationship to dataset documentation efforts. We'll address FAIR principles, data cards, NeurIPS guidelines, Becker et al. (2019), and Bhardwaj et al. (2024), which you pointed us to. Our write up will acknowledge these efforts' value in providing transparency and best practices for data documentation and curation processes. We'll further explain that benchmarks, which extend datasets with infrastructure and evaluation, require additional guidelines to support transparent decision-making of users and we will clarify how our paper thus builds upon and expands these existing data documentation efforts.
>
> Also, good point on the Croissant metadata record and datasheet! We intended to release both once we published the data but should have made this clear in the write up. We have included the datasheet and the croissant metadata (anonymized) in the attached PDF to credibly commit that these will be included upon publication :)

---

> > ### Author Response · Authors · 2024-08-27
> >
> > Dear reviewer fLHh, given that we’re more than halfway through the author-reviewer discussion period, we wanted to send a friendly reminder that we’d love to hear your thoughts on our rebuttal! Please let us know if there are any remaining questions or concerns. If we addressed your previous concerns sufficiently, we’d be delighted if you’d consider raising your final score. Thank you so much again for your time and feedback!

---

> > ### Comment · Reviewer_fLHh · 2024-08-29
> > **Response to author rebuttal**
> >
> > Thank you for your detailed reply to my comments. I think the additions discussed here in the rebuttal greatly improve the paper, and I look forward to seeing the updated results with the new scoring. I have raised my score.

---

> > > ### Author Response · Authors · 2024-09-01
> > > **Thanks!**
> > >
> > > Dear reviewer fLHh, thank you again for your constructive feedback – it genuinely improved our paper! We also thank you for raising your score. As an aside, based on preliminary findings, you were right, according to our new criteria, most benchmarks do not sufficiently detail the data curation process or use standardized documentation for their (meta-)data (if they include it at all). We’re excited for our new results, thanks again for your advice!

---

### Official Review · Reviewer_X3U4 · 2024-07-29
**Unclear goals and execution in the paper**

**Rating:** 4
**Confidence:** 5
**Correctness:** See section "Review"
**Clarity:** See section "Review"

**Review:**

> [4-5] They can inform model selection for downstream tasks and influence policy initiatives.

main contribution

> [7-9] In this paper, we develop an assessment framework considering 40 best practices across a benchmark's life cycle and evaluate 25 AI benchmarks against it.

quality differences like what?

> [10-11] We further find that most benchmarks do not report statistical significance of their results nor can results be easily replicated.

models using the benchmarks?

> [13-14] We also develop a living repository of benchmark assessments to support benchmark comparability.

what is your assessment?

> [19-20] AI evaluation is an essential disci-pline [12]. Current evaluation approaches include both internally developed procedures

are we talking about approaches or the evaluation itself?

> [23] Following the work of [62]

name the work in full

> [31-32] as one that is interpretable and clear about its intended purpose and scope, and that is usable

what does usable mean here?

> [33] date, no structured assessment for the quality of AI benchmarks, including both FM and non-FM

again, what does assessment mean here?

> [38] 16 foundation model (FM) and 9 non-FM benchmarks (full list in App. A), finding statistically significant quality differences within and across both categories.

I do not understand this - Why are benchmarks model specific? Aren't they usually model agnostic?

> [54-55] Regarding what a benchmark measures, [55] find that current benchmarks for LLMs are insufficient for assessing these models' complex capabilities.

I like this formulation of problem statement

> [56-62] Similarly, [57] finds that the rapid advancement of AI models threatens benchmarks' utility, as a large fraction of these evaluations are near saturation. [76] and [46] both address the narrow scope of existing benchmarks, with [46] advocating for approaches intended to reduce the socio-technical gap that exists between the capabilities that benchmarks are able to measure and the ability of models to meet human needs in downstream applications. With respect to how evaluations are used, [62] critiques the tendency of AI practitioners to overgeneralize benchmark results, highlighting how these scores present an inherently reductive view of model performance.

general comment about writing -> I think it is better to formulate it as -> researchers have found [55] or write the paper states "if you say 'XYZ' says."

> [72-73] Our work is informed by benchmarking practices from fields beyond AI, ranging from transistor hardware [15] to environmental quality [13] to bioinformatics [5], and identify common themes

While this is a good thing, why is this relevant to AI benchmarks? What are the commonalities?

> [109] concrete criteria

I am at page 3 and I still do not know what these criteria look like. Ideally, you would have examples of these 3 categories here.

> [118] members of all stakeholder groups

How many people?

> [129] We scored benchmarks on discrete 0/5/10/15 point scale for each criterion

This is a weird choice for a scoring scale instead of 1-4 or 0-3. The points here imply much more granularity than assessed.

> [Figure 2; points 2 and 10] How tested capability or concept translates to benchmark task is described -- and -- How knowing about the tested concept is helpful in the real world is described

are these different?

> [Figure 2; point 7] Has validated automatic evaluation

what does this imply?

> [Figure 2; point 8] Input sensitivity is addressed

which inputs? prompts or samples?

> [Figure 2; point 11] Informed performance metric choice

how is informed judgement used?

> [Figure 3; point 2] Evaluation data or generation mechanism is accessible

What does this mean?

> [Figure 3; point 6] Task to identify if model is included trained on benchmark data

data leakage is an unsolved problem though, what is the expectation here

> [171-173] Including a 'training_on_test_set' task allows determining whether a model's training data included benchmark examples, e.g., as used in [68].

the expectation is explained here -- but is unclear in the overview diagram

> [Figure 3; point 8] Statistical significance or uncertainty quantification of benchmark results is reported

That is the responsibility of person using on the benchmark not the one making it though?

> [Figure 4] In-line code comments are used

Why does it need a script, documentation and comments? There is an overlap here.

> [Figure 4; point 5] Accompanying paper is accepted at peer-reviewed venue

Why?

> [Figure 4; point 7] . Assumptions of **_normative properties_** are documented

What are these?

> [Figure 4; point 10] Data collection, test environment design, or general design process is documented

This should be in evaluation setup

> [181] normative assumptions

What does this mean or imply?

> [Figure 5] Maintenance Criteria

does this always need to be a living benchmark community?

> [188-191] An optimally designed, implemented, and documented benchmark will cease to be useful if it is not maintained. Developers should regularly check code usability and maintain a feedback channel for users to report issues or suggest improvements. Providing contact details of a person responsible for the benchmark facilitates communication and support.

needs more discussion about how resource intensive this expectation is.

> [200] can be

given how prompt sensitive current models are -> isn't this a wrong expectation?

> [208] task versioning

are you suggesting using semver?

> [218] Positionality statement. Positionality statements are a reflective account common in social sciences research. In them, researchers acknowledge how their background, experiences, and biases may have influenced their work. Developers believe such factors significantly impacted their benchmark's construction. If developers may provide a positionality statement for increased context and transparency.

I find this a pretty novel suggestion and interesting.

> [Figure 7] Foundation Model and Non-Foundation Model

This is the first time I am seeing the use of FM vs non-FM and that distinction does not make sense to me? All of these work with non-FM models too?

> [255-257] Most benchmarks fail to distinguish signal and noise. Benchmark developers should not only report a single result for a model but also re-run their evaluation [10] with, e.g., different random seeds or sampling temperatures, and report the mean and variance for these intra-model evaluations.

While this is great in theory, costs to do this is are pretty prohibitive in practice.

> [297-299] The scoring system differentiates only four categories to enable relatively objective evaluation through clear-cut criteria (App. J and App. D), but may miss nuances within each category.

Why choose such a high-fidelity rating scale for this purpose if fuzziness is not allowed in score?

**Strengths:**

See section "Review"

**Additional Feedback:**

N/A

**Documentation:**

While I don't think this requires particular documentation, they do have a  significant amount of documentation in the appendix though and I think that is  that is in general fine.

**Limitations:**

See section "Review"

**Opportunities For Improvement:**

See section "Review"

**Relation To Prior Work:**

See section "Review"

**Summary And Contributions:**

The paper develops an assessment framework comprising 40 best practices across a benchmark's life cycle to evaluate 25 AI benchmarks. The authors claim to find significant quality differences among these benchmarks and note that they often fail to report statistical significance. However, the paper's approach is confusing, as it conflates benchmarks with their associated leaderboards and doesn't clearly differentiate between the benchmark datasets and the reported results. The paper sets unrealistically high standards, limiting its practical utility. Additionally, it attempts to address both benchmark creation and deployment/usage/reporting without clearly distinguishing between these aspects -- and working on the interlink between the two, which diminishes its overall value.

---

> ### Author Rebuttal · Authors · 2024-08-17
>
> Thank you for your review! We believe that there have been some misconceptions about our work and wanted to clarify a few seemingly misunderstandings and add information where you raised questions:
>
> **Response to Summary Section:**
>
> - You mentioned _"Unclear goal"_ in your title, but your review doesn't address this. Could you please clarify what was unclear? Our aim is to establish a minimum standard for AI benchmarks and provide a checklist for developers, as these didn't previously exist in the community. The living directory is intended to help users identify benchmarks adhering to these best practices. More broadly, we aim to foster transparency and an improvement of AI benchmark quality with our work. Within the small circles we shared the draft paper, we already received very positive feedback from benchmark developers and users who have referred to the guidelines as part of their work, which makes us optimistic that this can also be a valuable resource for the broader community.
>
> - _“However, the paper's approach is confusing, as it conflates benchmarks with their associated leaderboards”_ →
> We never mention leaderboards in the paper or in the context of our assessment. Could you elaborate your criticism here? We also couldn’t find a related comment in your detailed review.
>
> - _“Doesn't clearly differentiate between the benchmark datasets and the reported results”_ →
> It seems there's been a misunderstanding regarding the “statistical significance of benchmark results” that you mention in your review, which seems to be driving this feedback. We're not referring to results that model developers run. Rather, we mean the statistical significance that benchmark developers should report for their initial results with the models they initially test. This information is relevant for benchmark users to know if a benchmark is actually capturing a signal. This is also a requirement of NeurIPS as part of the mandatory checklist for authors submitting any experiments (e.g., running a new benchmark on a model).
> - _“It attempts to address both benchmark creation and deployment/usage/reporting without clearly distinguishing between these aspects”_ → All of our criteria are only meant to pertain to benchmark developers, they do not present criteria for benchmark users or consumers. Can you elaborate what you mean here?
>
> - _“The paper sets unrealistically high standards”_ →
> We address this feedback from you below, which pertains to only a small number of our criteria (2 out of 40). Given our own experience with developing benchmarks, we strongly believe that these two (feedback channel & reporting statistical significance) don’t present unrealistic asks for benchmark developers. We believe that we should hold benchmarks to a high standard, given their impact and use across academia, industry, and policy. The lack of standards and reference guides in the benchmark community has resulted in the publication of many benchmarks that are not usable at best and misleading to benchmark users at worst.
>
> **Response to Detailed Review:**
> - _[72-73]:_ Understanding how constructs such as benchmarks are being used beyond AI can help to derive better practices for the AI community, since benchmarks have been around much longer (and hence matured) in other fields. Works in other fields have for example requirements for transparency and insights about benchmark design and purpose that we adapted for our AI benchmark assessment.
> - _[Figure 2; points 2 and 10], [Figure 2; point 7], [Figure 2; point 7], [Figure 2; point 8], [Figure 2; point 11], [Figure 3; point 2], [Figure 3; point 6], [Figure 4; point 5], [Figure 4; point 7]_: You can find detailed explanation and justification for each criteria in Appendix J. Please let us know if the information there answers your questions; if not, we’d be happy to provide additional details and add clarifications to the write up!
> - _[Figure 4]:_ The script is for a benchmark user to run and replicate the exact evaluation the benchmark developers used. Separate code documentation provides a high-level overview and explanation of the software's structure, functionality, and usage, while inline comments offer immediate, context-specific clarifications and explanations within the code itself. This is common practice in software engineering with well-established standards (see, e.g., IEEE 1028, PEP8, PEP257), and hence benchmarks should adhere to similar principles.
> - _[188-191]:_ It’s unclear to us what you mean by “living benchmark community.” Our maintenance criteria include a feedback channel, a listed contact person, and recent code usability checks. Could you please clarify your comment? Regarding resource intensity, the minimum requirement is one person and an active Github directory with maintained issues. We only require responses to issues, not resolution. For context, prominent benchmarks like BIGBench receive about 2-5 new issues monthly, which we believe is manageable for one individual to respond to. Alternatively, as outlined in our paper, benchmarks can be retired and marked as not actively maintained. In this case, they would be labeled 'retired' in our directory and not scored in the maintenance category.
> - _[38]:_ Benchmarks are model-agnostic within their designed scope, but not universally applicable. For instance, MMLU tests foundation models' pre-trained knowledge, which wouldn't apply to a specialized image classification model. We introduced this distinction for analysis purposes, highlighting the difference between general-capability and task-specific benchmarks. However, our proposed criteria are designed to apply broadly to all AI benchmarks, with few exceptions. This distinction doesn't detract from our main contributions or findings; it merely provides context for our analysis.

---

> > ### Author Rebuttal · Authors · 2024-08-17
> >
> > - _[255-257]:_ Most benchmarks we’ve seen (and built ourselves) aren’t costly to run on different models (they are costly to build). The expectation here is that they rerun the benchmark at least twice (better more) for each model they initially report and report the statistical significance of the results, which, given the current cost of running a benchmark, seems like a reasonable expectation in our perspective. This is also in line with NeurIPS requirements which require reporting of statistical significance analysis as part of the required checklist for the main conference and D&B track. These requirements have further been iteratively developed with members of the benchmark community (see Methodology & clarifications to reviewer fLHh), suggesting that there is some agreement that these don't present unreasonable standards.
> > - _[129]:_ Previous research has shown that wider linear scales are more easily interpretable by humans. Given that both scales (0, 1, 2, 3) and (0, 5, 10, 15) are linear, they do not change our analysis results. However, we’re open to changing the scaling in our paper pre-publication if this is a key concern.
> >
> > **Clarifications necessary from you**
> > - Responses to open questions pertaining to the summary section that we outlined in the Section "Response to Summary Section"
> > - _[7-9]:_ We’re not sure what you mean here, can you elaborate?
> > - _[13-14]:_ We are not sure what you mean here, the assessment is described in detail in the paper and in line [7-9] we clarify that the assessment consists of 40 best practices we identified. Could you clarify what you think is missing here here?
> > - _[23]:_ We do not understand your comment here, we are using the standard Neurips citation guidelines. Could you please elaborate?
> > - _[33]:_ We don’t understand your question here, could you please clarify? To what extent is the term “assessment” unclear in this sentence and would this be resolved if we added example criteria in the introduction, as you suggested in your review?
> > - _[Figure 4; point 10]:_ Could you provide your reasoning here for why this should be under evaluation setup?
> >
> > Based on your suggestions, we’ll add examples of criteria to the introduction, and add the above-provided clarifications to the paper and we’d be happy to address concerns regarding the seemingly misunderstandings once we better understand your specific concerns.
> >
> > Please let us know if there are any follow-up concerns or questions. Given that there seem to have been a variety of misunderstandings of the paper and some of your previous open questions can be answered with information is already available in the paper as part of the appendix (which you weren't required to read, of course, in the first review of the paper!), we kindly ask you to revise your score in light of the above provided clarifications. Thank you again for your time and feedback!

---

> > > ### Author Response · Authors · 2024-08-27
> > >
> > > Dear reviewer X3U4, given that we’re more than halfway through the author-reviewer discussion period, we wanted to send a friendly reminder that we’d love to hear your thoughts on our rebuttal! Please let us know if there are any remaining questions or concerns. If we addressed your previous concerns sufficiently, we’d be delighted if you’d consider raising your final score. Thank you so much again for your time and feedback!

---

### Author Response · Authors · 2024-08-17
**Summary of Author/Reviewer Discussions**

**Summary of our work:**

Our research is the first of its kind that identifies, with input from 15+ stakeholders, 40 best practices for AI benchmark developers which we use as criteria to score existing AI benchmarks. We find significant quality differences and issues in popular benchmarks. We provide actionable guidelines for benchmark developers to adhere to these best practices and improve the quality of benchmarks in the field.

**Contributions As Seen By the Reviewers:**

We appreciate that reviewers see that our research _"presents high-quality work in assessing AI/ML benchmark practices"_ (reviewer flHh) and that our checklist _“has the potential to be a very useful tool for the ML community”_ (reviewer fLHh). They further find that _"this paper provides significant contributions to build and assess usable benchmarks"_ (reviewer 2Wd9) and that we present "a comprehensive approach to evaluating benchmarks across [the] AI life cycle" (reviewer 2Wd9). Reviewer fLHh further highlights that we _"did not only consult with AI/ML researchers during their development process"_ ensuring "diversity of perspectives on the considerations of downstream users". They further note that we _"clearly explain their methodology and rubric through excellent use of figures"_.

**Improvements Made Based on Reviewer Feedback:**

In response to the reviewers' suggestions, we have implemented the following major improvements to our work:
- Following a recommendation of reviewer fLHh, we clarified the iterative development process of both the benchmark lifecycle and criteria development. We have put together a flow diagram of our process and added it to the appendix of the paper.
- Following the recommendation of reviewer fLHh, we're adding eight new data curation criteria to our documentation requirements. We're rescoring benchmarks accordingly and will update results pre-publication. This is in conflict with reviewer X3U4’s notion that even before this addition, they noted that we were setting _"unrealistically high standards"_. However, we concur with reviewer flHh that these criteria are necessary to ensure high benchmark quality (for a detailed discussion, see section below and our rebuttal to reviewer X3U4).
- Following a recommendation of reviewer X3U4, we added example criteria to the introduction.
- Following an open question from reviewer X3U4, we clarified that the reporting of statistical significance pertains to the benchmark developers in the context of our assessment, which is in line with NeurIPS guidelines to report statistical significance.
- Following a recommendation of reviewer fLHh, we added a discussion to our prior work section explaining how our guidelines extend and complement existing dataset guidelines and standards.
- Following a recommendation of reviewer fLHh, we added Croissant metadata and a data sheet for our assessment dataset.
- Following a recommendation of reviewer 2Wd9, we explicitly called for benchmark developers to use our guidelines as a minimum quality standard.
- Following a recommendation of reviewer 2Wd9, we expanded the future work section of our work.
Reviewer fLHh notes that the _"additions discussed here in the rebuttal greatly improve the paper"_.

If our research is accepted, all updates will be integrated in the write-up, using the extra page limit for the camera-ready paper.

**Other Comments on Reviewer Feedback:**

- We respectfully disagree with reviewer X3U4's notion that our criteria present _"unrealistically high standards"_. According to their review, this feedback only pertains to 2 out of 40 (now 48) criteria. We further respectfully disagree that these two criteria (reporting statistical significance and maintaining a feedback channel or retiring the benchmark) present too high of a bar for benchmark developers. We consulted benchmark developers as part of our iterative development process and all of them agreed that these are reasonable criteria; we have further found that these criteria have been successfully adopted in previous AI benchmarks.
- As outlined in our rebuttal to reviewer X3U4, we find that the reviewer missed relevant information in the paper. While we answered their questions and pointed them to the missed information, they never responded to us or changed their score during the rebuttal period. There were also multiple instances were we asked for clarifications from reviewer X3U4 since their feedback was unclear (e.g., they mention that our paper _“conflates benchmarks with their associated leaderboards”_ when we never mention leaderboards); these were also left unanswered. Hence, we were unfortunately not able to take large parts of reviewer X3U4’s feedback into account.

**Thank You:**

We are grateful to all reviewers for their insightful comments and constructive feedback. We further would like to thank all reviewers, ACs, SACs, and PCs for their time and hard work, especially with record submission numbers to NeurIPS this year.

---

### Decision · Program_Chairs · 2024-09-26

**Decision:**

Accept (Spotlight)

**Comment:**

Meta review of Betterbench: Assessing AI Benchmarks, Uncovering Issues, and Establishing Best Practices
NeurIPS 2024 Datasets and Benchmarks Track Submission742

After assessing the paper and the reviews, I recommend that this paper be accepted as a spotlight poster. While the average rating is 6.0 and the reviews are altogether controversial (with two scores of 7.0 and one of 4.0) there are several reasons for this paper to be accepted.

First, the AC’s have been instructed to seek out papers that provide the most interesting and engaging program for NeurIPS. This paper is well-positioned to provoke engaging discussion at the conference, more so than other papers in this track. Its topic material is the relative quality of benchmarks themselves. While the field of benchmarking is growing crowded, this paper takes a different tack by investigating how we might assess the quality and usability of benchmarks. As the authors point out, there is currently no way to assess the quality of benchmarks. If the benchmark is to endure as a useful and widely applicable tool, there must be some way to report the relative quality of such tools to potential adopters across industry, academic, and policy. This paper does that work, by contributing a novel framework for such assessment, and using empirical interviews to assess usability across different personas of users. Further, by putting this framework into practice, the authors identify serious issues which ought to be platformed at NeurIPS, including poor replicability. Again, if the benchmark is to endure as a tool that is taken seriously, then we as a community must confront issues in quality and replicability (an issue which has undercut many academic sciences). As Reviewer fLHh pointed out, the framework developed in this paper “has the potential to be a very useful tool for the ML community” and Reviewer 2Wd9 noted "this paper provides significant contributions to build and assess usable benchmarks,” which is valuable for the majority of the Datasets & Benchmarks community. Further, such contributions make the Neurips Datasets & Benchmarks track a very good fit and venue for this paper, where it can do the most to contribute to state of the art.

Secondly, I believe that the two higher ratings are more accurate in their representation of quality than the lower rating. During the discussion period, authors pointed out some inaccuracies and points of confusion in the review associated with the lower rating. Upon review I agree with the authors’ assessment, and argue that this paper should be accepted despite this lower rating.

Third, the authors responded well to the reviewers’ recommendations. For example, Reviewer fLHh made excellent points about the lack of data documentation, such as the inclusion of FAIR principles and datacards. The authors responded by agreeing with the improvement such a revision would make to their paper, and have agreed to implement this revision in their camera-ready.

In conclusion, I recommend Accept (Spotlight).